# Electrical synaptic transmission requires a postsynaptic scaffolding protein

Abagael M Lasseigne[1†], Fabio A Echeverry[2†], Sundas Ijaz[2†], Jennifer Carlisle Michel[1†], E Anne Martin[1], Audrey J Marsh[1], Elisa Trujillo[1], Kurt C Marsden[3], Alberto E Pereda[2]*, Adam C Miller[1]*

[1]Institute of Neuroscience, University of Oregon, Eugene, United States; [2]Dominick P. Purpura Department of Neuroscience, Albert Einstein College of Medicine, Bronx, United States; [3]Department of Biological Sciences, NC State University, Raleigh, United States

**Abstract** Electrical synaptic transmission relies on neuronal gap junctions containing channels constructed by Connexins. While at chemical synapses neurotransmitter-gated ion channels are critically supported by scaffolding proteins, it is unknown if channels at electrical synapses require similar scaffold support. Here, we investigated the functional relationship between neuronal Connexins and Zonula Occludens 1 (ZO1), an intracellular scaffolding protein localized to electrical synapses. Using model electrical synapses in zebrafish Mauthner cells, we demonstrated that ZO1 is required for robust synaptic Connexin localization, but Connexins are dispensable for ZO1 localization. Disrupting this hierarchical ZO1/Connexin relationship abolishes electrical transmission and disrupts Mauthner cell-initiated escape responses. We found that ZO1 is asymmetrically localized exclusively postsynaptically at neuronal contacts where it functions to assemble intercellular channels. Thus, forming functional neuronal gap junctions requires a postsynaptic scaffolding protein. The critical function of a scaffolding molecule reveals an unanticipated complexity of molecular and functional organization at electrical synapses.

*For correspondence:
alberto.pereda@einsteinmed.org (AEP);
acmiller@uoregon.edu (ACM)

†These authors contributed equally to this work

Competing interests: The authors declare that no competing interests exist.

## Introduction

Synapses are specialized cellular adhesions between neurons that rapidly transfer information to facilitate neural network function. There are two modalities of fast synaptic transmission, chemical and electrical, both found throughout animal nervous systems including in mammals (*Moroz and Kohn, 2016*; *Ryan and Grant, 2009*). Chemical synapses are inherently asymmetric structures, derived from presynaptic specializations that regulate the release of neurotransmitters and postsynaptic specializations that contain neurotransmitter receptors and the machinery required to propagate signal transmission. Both specializations require hundreds to thousands of proteins, which together tightly control the structure, function, and modulation of synaptic communication (*Ackermann et al., 2015*; *Grant, 2019*; *Siddiqui and Craig, 2011*). For example, intracellular scaffolding proteins of the Post Synaptic Density (PSD) at chemical synapses regulate the number and functional state of AMPA and NMDA receptors, which are ligand-gated ion channels, at glutamatergic synapses (*Zhu et al., 2016*). By contrast, electrical synapses are often perceived as simple aggregates of intercellular channels known as gap junctions (GJs) (*Goodenough and Paul, 2009*). Intercellular GJ channels are formed by the docking of two hemichannels, composed of Connexin proteins in vertebrates and Innexins in invertebrates (*Bhattacharya et al., 2019*; *Phelan, 2005*; *Shruti et al., 2014*; *Söhl et al., 2005*). Each neuron contributes a hemichannel from each side of the synapse, which form a channel and support communication by allowing the spread of electrical currents and small metabolites between adjacent 'coupled' neurons. While multiple Connexins and Innexins can contribute to individual electrical synapses (*Bhattacharya et al., 2019*; *Miller et al.,*

**eLife digest** Neurons 'talk' with each another at junctions called synapses, which can either be chemical or electrical. Communication across a chemical synapse involves a 'sending' neuron releasing chemicals that diffuse between the cells and subsequently bind to specialized receptors on the receiving neuron. These complex junctions involve a large number of well-studied molecular actors.

Electrical synapses, on the other hand, are believed to be simpler. There, neurons are physically connected via channels formed of 'connexin' proteins, which allow electrically charged ions to flow between the cells. However, it is likely that other proteins help to create these structures. In particular, recent evidence shows that without a structurally supporting 'scaffolding' protein called ZO1, electrical synapses cannot form in the brain of a tiny freshwater fish known as zebrafish. As their name implies, scaffolding proteins help cells organize their internal structure, for example by anchoring other molecules to the cell membrane.

By studying electrical synapses in zebrafish, Lasseigne, Echeverry, Ijaz, Michel et al. now show that these structures are more complex than previously assumed. In particular, the experiments reveal that ZO1 proteins are only present on one side of electrical synapses; despite their deceptively symmetrical anatomical organization, these junctions can be asymmetric, like their chemical cousins. The results also show that ZO1 must be present for connexins to gather at electrical synapses, whereas the converse is not true. This suggests that when a new electrical synapse forms, ZO1 moves into position first: it then recruits or stabilizes connexins to form the channels connecting the two cells.

In many animals with a spine, electrical synapses account for about 20% of all neural junctions. Understanding how these structures form and work could help to find new treatments for disorders linked to impaired electrical synapses, such as epilepsy.

*2017*; *Phelan et al., 2008*; *Rash et al., 2013*), the complexity of neuronal GJ cellular biology (*Lynn et al., 2012*; *Martin et al., 2020*; *Meyer et al., 2014*; *Sigulinsky et al., 2020*) and the variety of mechanisms regulating their synaptic strength (*Arroyo et al., 2016*; *Bloomfield and Völgyi, 2009*; *Marder, 1998*; *O'Brien and Bloomfield, 2018*; *Pereda, 2014*) suggest they require complex multimolecular structures to support function.

Several Connexin-associated proteins have been identified (*Lynn et al., 2012*; *Miller and Pereda, 2017*); however, it remains undetermined whether such associated proteins are ancillary to the channels or requisite for electrical synapse function. Perhaps the best characterized Connexin-associated protein is Zonula Occludens 1 (ZO1) (*Bauer et al., 2010*; *Willott et al., 1993*), which is an intracellular scaffolding protein and a member of the membrane-associated guanylate kinase (MAGUK) family of proteins. MAGUKs constitute a large family of multifunctional adaptor proteins that play key roles in scaffolding membrane channels and receptors to intracellular signaling complexes and the cytoskeleton (*González-Mariscal et al., 2000*). MAGUK proteins, including ZO1, contain PSD95/Dlg/ZO1 (PDZ) protein-protein interaction domains, which bind to PDZ-binding motifs often located at the carboxy terminus of partner proteins, including Connexins (*Zhu et al., 2016*). The best studied ZO1/Connexin interaction is with Connexin 43 (Cx43), a widely expressed, non-neuronal, GJ-channel forming protein (*Giepmans and Moolenaar, 1998*). The ZO1/Cx43 interaction is thought to play important functional roles in GJ regulation by facilitating the docking of newly inserted hemichannels, which promotes the formation of intercellular channels (*Hunter et al., 2005*). Moreover, the ZO1/Cx43 interaction is critical for channels to remain conductive prior to removal during channel turnover at epithelial GJs (*Hervé et al., 2014*; *Thévenin et al., 2017*). While ZO1 is an important regulator of Cx43-contaning GJs, less is known about its role at neuronal GJs, which are primarily formed by the Cx36-family of proteins and mediate electrical synaptic transmission in vertebrate nervous systems (*Connors and Long, 2004*; *Miller et al., 2017*; *Rash et al., 2013*; *Söhl and Willecke, 2004*). In neurons, ZO1 immunostaining correlates with synapses containing Cx36 and its fish homologs (*Flores et al., 2008*; *Li et al., 2004*; *Marsh et al., 2017*; *Yao et al., 2014*), and its presence at synapses may play regulatory roles (*Flores et al., 2008*), the nature of its contributions to electrical transmission remains unknown.

Despite mounting evidence for the widespread dynamic functional contributions of electrical synapses to neural circuit function, the perception of the simplicity of their molecular organization remains. We hypothesized that scaffolding molecules form part of a multimolecular structure that is required for channel function akin to that found at chemical synapses. Here, we explore the functional role of ZO1 in zebrafish by examining identifiable synaptic contacts of the Mauthner cell (*Bartelmez, 1915*; *Bodian, 1937*; *Hildebrand et al., 2017*; *Kimmel, 1982*; *Robertson et al., 1963*), which forms stereotyped electrical synapses accessible to genetic, biochemical, cell biological, electrophysiological, and behavioral analyses. We show that the presence ZO1 protein is critically required for the structure and function of the intercellular channels. Moreover, we find that the localization of ZO1 is compartmentalized postsynaptically where it functions in the formation of neuronal GJs. Our results stand in contrast with current views on electrical synapse organization centered solely on the proteins forming GJ channels. Thus, our findings provide strong support to the notion that electrical synapses constitute complex and asymmetric synaptic structures at which intercellular channels are governed by multimolecular structures with features that parallel the molecular and functional organization of the PSD at chemical synapses.

## Results

### ZO1b is required for robust Connexin localization to electrical synapses

We sought to examine the relationship between the intracellular scaffold ZO1 and neuronal Connexins (Cxs) by utilizing the stereotyped synapses of the zebrafish Mauthner cell. This circuit drives a fast escape response to threatening stimuli using both electrical and chemical connections (*Eaton et al., 1977*; *Jacoby and Kimmel, 1982*; *Liu and Fetcho, 1999*; *Wolman et al., 2015*). Each animal has two Mauthner cells that receive multimodal sensory input that relay information to the spinal cord to coordinate circuits to elicit fast turns. We focus on two populations of stereotyped electrical synapses made by Mauthner cells: (1) 'club ending' (CE) synapses (*Bartelmez and Hoerr, 1933*; *Pereda et al., 2004*; *Yao et al., 2014*), which are mixed electrical/glutamatergic chemical synaptic contacts formed between auditory afferents of the eighth cranial nerve and the Mauthner cell's lateral dendrite (*Figure 1A,B*) and (2) *en passant* electrical synapses formed between the Mauthner axon and Commissural Local (CoLo) interneurons found in each spinal-cord segment (*Figure 1A,M*/CoLo synapses) (*Satou et al., 2009*). Neuronal GJs at both CEs and M/CoLo synapses are made of heterotypic channels formed by Cx35.5, encoded by the gene *gap junction delta 2a* (*gjd2a*), and Cx34.1, encoded by *gjd1a*, both homologous to mammalian Cx36 (*gjd2*). We previously found that Cx35.5 and Cx34.1 are localized asymmetrically to the pre- and postsynaptic sides at CE and M/CoLo synaptic contacts (*Figure 1B*; *Miller et al., 2017*). Throughout we use the terms pre- and postsynaptic to reference the neuronal, cell-biological compartment in which a Connexin is localized. At CE synapses, the auditory afferent axons are presynaptic to the postsynaptic Mauthner lateral dendrite; while at M/CoLo synapses, the Mauthner axon is presynaptic to the postsynaptic CoLo.

We first determined the localization of ZO1 and the Connexin proteins at Mauthner electrical synapses using immunofluorescence and confocal imaging. We stained 5 day post fertilization (dpf) larvae, a time at which the Mauthner circuit elicits a mature startle response, with antibodies against the human ZO1 protein and those that distinguish the zebrafish Cx35.5 and Cx34.1 (*Figure 1C–L*; *Miller et al., 2017*). We observed extensive colocalized signal for these three proteins at CE and M/Colo electrical synapses, with each protein apparent in the stereotyped shape and position of the neuronal gap junctions (GJs) at these contacts. We identified CEs unambiguously as large (1.5–2 μm) oval areas localized in the distal portion of the lateral dendrite of the Mauthner cell (*Figure 1C*; *Yao et al., 2014*). M/Colo synapses were identified by their regularly spaced sites of contact in the spinal cord (*Figure 1K*). Next, we examined the role of ZO1 at electrical synapses using CRISPR/Cas9-induced mutations to knock out gene function. Mammalian ZO1 is encoded by the gene *tight junction protein 1* (*tjp1*), while zebrafish have two homologous genes, *tjp1a* and *tjp1b*. Using a CRISPR-based screen, we found that mutations in *tjp1b/ZO1b*, but not *tjp1a/ZO1a*, caused a failure of Connexin localization at M/CoLo synapses (*Figure 1*; *Figure 1—figure supplement 1*; *Marsh et al., 2017*; *Shah et al., 2015*). We examined the effect of these mutations on CEs and found that *tjp1b/ZO1b*$^{-/-}$ mutants lack most of the detectable fluorescent staining for both Cx35.5 and Cx34.1, as well as ZO1, at the stereotyped synaptic contact sites (*Figure 1D,L*).

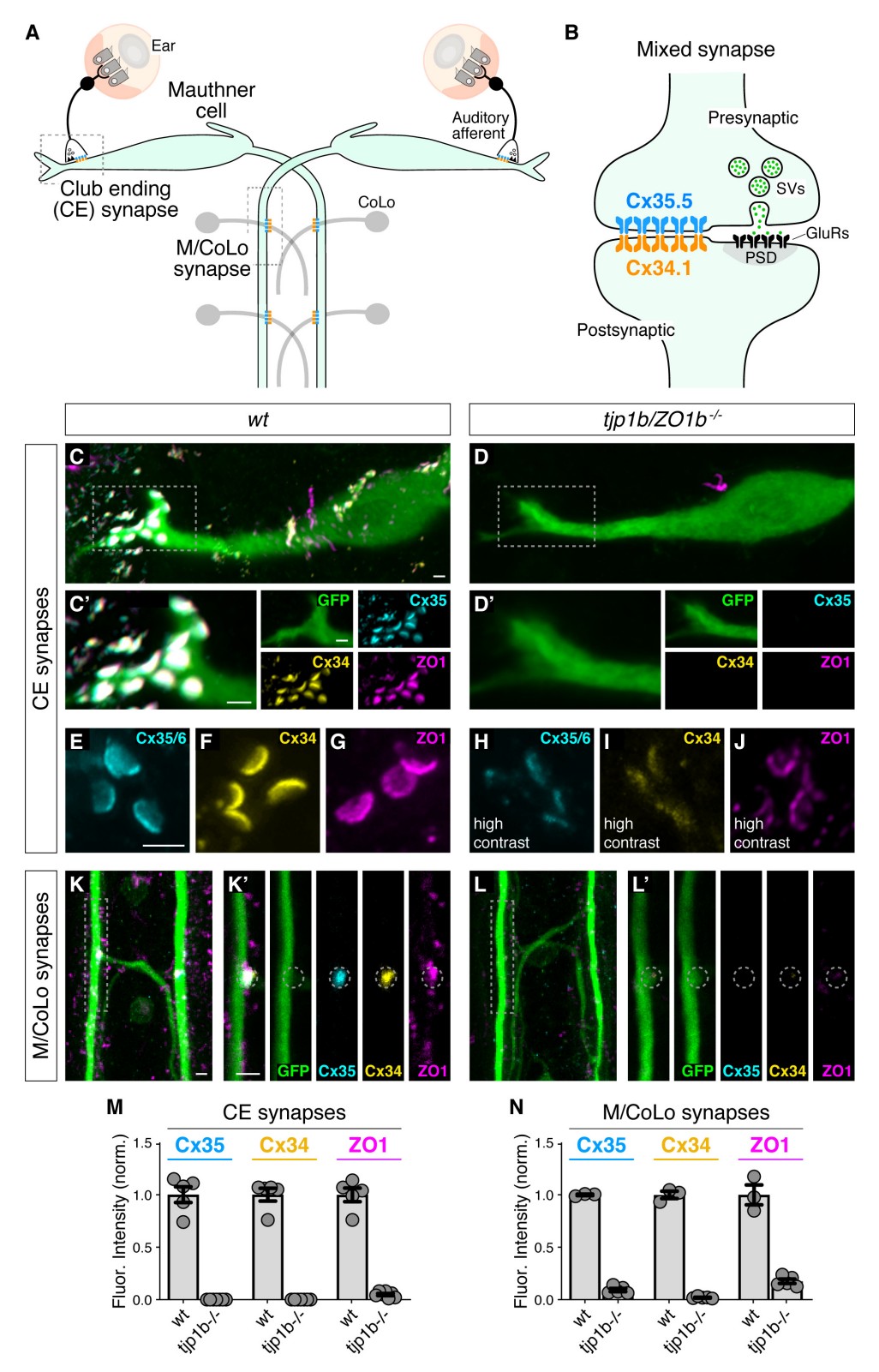

**Figure 1.** Localizing Connexin to electrical synapse contacts requires the intracellular scaffold protein ZO1b. (**A**) Simplified diagram of the Mauthner cell circuit illustrating the electrical synapses of interest. The image represents a dorsal view with anterior on the top. Boxed regions indicate regions stereotypical synaptic contacts used for analysis. Presynaptic auditory afferents contact the postsynaptic Mauthner cell lateral dendrite in the hindbrain forming mixed electrical/glutamatergic Club Ending (CE) synapses. In the spinal cord, the presynaptic Mauthner axons form *en passant* electrical

*Figure 1 continued on next page*

*Figure 1 continued*

synapses with the postsynaptic CoLo interneurons (M/CoLo synapses) in each spinal cord hemisegment (2 of 30 repeating spinal segments are depicted). Electrical synapses are denoted as rectangles depicting the two Connexin (Cx) hemichannels (presynaptic Cx35.5 [cyan] and postsynaptic Cx34.1 [yellow]) that form the neuronal gap junction channels of this circuit. (B) Diagram of a mixed electrical/glutamatergic synapse as found at CEs. In the electrical component, molecularly asymmetric Connexin hemichannels (Cx35.5 [cyan], Cx34.1 (yellow)) directly couple cells. In the chemical component, presynaptic synaptic vesicles (SVs) release neurotransmitter (green circles) which align with postsynaptic glutamate receptors (GluRs). The formation and function of chemical synapses are regulated by scaffolds of the postsynaptic density (PSD, gray). (C–L) Confocal images of Mauthner circuit neurons and stereotypical electrical synapse contacts in 5-day-post-fertilization, *zf206Et*, transgenic zebrafish from *wildtype* (*wt*, C,E–G,K) and *tjp1b/ZO1b*$^{-/-}$ mutant animals (D,H–J,L). In panels (C,D,K,L) animals are stained with anti-GFP (green), anti-zebrafish-Cx35.5 (cyan), anti-zebrafish-Cx34.1 (yellow), and anti-human-ZO1 (magenta). In panels (E–J), animals are stained individually with the indicated antibody. Scale bar = 2 µm in all images. (C, D) Images of the Mauthner cell body and lateral dendrite in the hindbrain. Images are maximum intensity projections of ~15 µm. Boxes denote location of CE contact sites and this region is enlarged in C′ and D′. In C′ and D′ images are maximum-intensity projections of ~5 µm and neighboring panels show individual channels. (E–J) Images of the Mauthner CEs stained for individual electrical synapse components. Images are maximum-intensity projections of ~3.5 µm. In the *tjp1b/ZO1b*$^{-/-}$ mutant panels (H–J), the contrast for each channel was increased in order to visualize the staining that remained at the synapses. (K,L) Images of the Mauthner/CoLo processes and sites of contact in the spinal cord. Images are maximum-intensity projections of ~5 µm. Boxes denote regions enlarged in K′ and L′. In K′ and L′ images are individual Z-sections and neighboring panels show individual channels. (M,N) Quantification of Cx35.5 (cyan), Cx34.1 (yellow), and ZO1 (magenta) fluorescence intensities at CE (M) and M/CoLo (N) synapses for the noted genotypes. The height of the bar represents the mean of the sampled data normalized to the *wt* average, and circles represent the normalized value of each individual animal (CE synapses: *wt* n = 5, *tjp1b/ZO1b*$^{-/-}$ n = 7; M/CoLo synapses: *wt* n = 3, *tjp1b/ZO1b*$^{-/-}$ n = 5). Error bars are ± SEM. For each comparison, *wt* and *tjp1b/ZO1b*$^{-/-}$ values are significantly different (Welch's t-test, p<0.01). Associated experimental statistics can be found in *Figure 1—source data 1*.

The online version of this article includes the following source data and figure supplement(s) for figure 1:

**Source data 1.** Source data for *Figure 1*.
**Figure supplement 1.** Characterization of ZO1 and Connexin mutants.
**Figure supplement 1—source data 1.** Source data for *Figure 1—figure supplement 1*.

Quantitation of Cx35.5, Cx34.1, and ZO1 fluorescence at CEs and M/CoLo contacts confirmed that staining for all three antibodies was greatly diminished in *tjp1b/ZO1b*$^{-/-}$ mutants (*Figure 1M,N*). By contrast, homozygous *tjp1a/ZO1a*$^{-/-}$ mutants had extensive Connexin and ZO1 staining at these contacts, while *tjp1a*$^{-/-}$; *tjp1b*$^{-/-}$ double mutants were indistinguishable from *tjp1b*$^{-/-}$ single mutants (*Figure 1—figure supplement 1A–G*). We conclude that ZO1b protein, encoded by the *tjp1b* gene, is localized to electrical synapses and required for the robust localization of both Cx35.5 and Cx34.1 at synaptic contacts.

While *tjp1b/ZO1b*$^{-/-}$ mutants showed significantly diminished levels of ZO1 and Connexin staining at presumptive synaptic locations, we wondered whether neurons were still attempting to assemble GJs. Indeed, we detected both Cx35.5 and Cx34.1 by western blot from brain homogenates of *tjp1b/ZO1b*$^{-/-}$ mutant animals (*Figure 1—figure supplement 1H*). We therefore examined CEs using higher contrast and magnification to assess GJ structure as detectable by immunolabeling and individually stained for ZO1 or Connexin to avoid confounding the image analysis due to bleed through of signal amongst stained proteins. We found that in *tjp1b/ZO1b*$^{-/-}$ mutants, each of the three antibodies revealed structures located at the stereotyped position of CE contacts and had morphologies reminiscent of wild-type animals, albeit with much dimmer fluorescence intensity (*Figure 1E–J*; note that image contrast was increased in mutants (H-J)). While we observed the stereotypical oval-shaped CE structures in mutants, the staining for each protein was weak and irregular in its distribution, suggesting the residual staining in mutants might represent incomplete, abortive synaptic structures (see electrophysiology below). Consistent with a reduced presence of GJ proteins, we also observed a reduced number of CEs detected by immunolabeling in *tjp1b/ZO1b*$^{-/-}$ mutants (*Figure 1—figure supplement 1I*). We found similar results at M/CoLo synapses, although their smaller size precluded an analogous detailed analysis (*Figure 1K,L,N*). In contrast to *tjp1b/ZO1b*$^{-/-}$, the staining of *tjp1a/ZO1a*$^{-/-}$ mutants was indistinguishable from wildtype (*Figure 1—figure supplement 1E–G,I*). These observations suggest that neurons of the Mauthner cell network in *tjp1b/ZO1b*$^{-/-}$ mutants persist in attempting to create electrical synapses, despite their inability to robustly localize neuronal Connexins at synaptic contacts. We conclude that ZO1 is localized to electrical synapses where it plays a critical role in neuronal GJ formation.

## ZO1b can localize to the electrical synapse independent of Connexins

Given that Connexin localization was dependent on ZO1b, we sought to determine if the converse was true – did ZO1 localization require Connexins? Using previously generated mutations in *gjd2a/Cx35.5* and *gjd1a/Cx34.1* (*Miller et al., 2017*), we examined the localization of Connexin and ZO1 proteins in mutants by immunolabeling (*Figure 2A–L*). First, we found that *gjd2a/Cx35.5*$^{-/-}$ and *gjd1a/Cx34.1*$^{-/-}$ mutant animals revealed a complete loss of detectable staining for the mutated protein. In addition, there was a failure of the non-mutated Connexin protein to robustly localize to the electrical synapse although low levels of staining were present. In line with these observations, by using brain homogenates and western blots, we found a complete loss of the Connexin affected by each mutation, but no effect on the non-mutated Connexin protein (*Figure 1—figure supplement 1H*). By contrast, ZO1 staining in the Connexin mutants was robust at the synaptic contact sites with the stereotyped appearance, distribution, and position clearly evident (*Figure 2A–L*). By comparing the relative ZO1 fluorescence between wild-type and Connexin mutant animals, we found that ZO1 was present at synaptic contacts at approximately half the normal level (*Figure 2M,N*). We examined *gjd2a*$^{-/-}$; *gjd1a*$^{-/-}$ double mutants and found that ZO1 still robustly localized to CE and M/CoLo contact sites (*Figure 2M,N*; *Figure 2—figure supplement 1A–F*). These results reveal two critical organizational principles about electrical synapses in the Mauthner cell: (1) ZO1b localizes to putative electrical synaptic sites largely independent of Connexin proteins and (2) each Connexin requires the other for robust localization to the synapse. Based on these data, we conclude that ZO1 can localize to neuronal GJs independent of the presence of channel-forming proteins, yet ZO1 is absolutely essential for proper Connexin localization.

To further examine the electrical synapse structure of Connexin mutants, we assessed CEs at higher contrast (*Figure 2D–I*). We found that: (1) immunofluorescence for the mutated Connexin is completely lost at the synapse, (2) the non-mutated Connexin is detectable but with weak and irregular labeling, suggestive of incomplete, abortive structures, and (3) ZO1 labeling resembles wildtype with a distribution and morphology that appears normal across the expanse of the putative synaptic contact. Consistent with these findings, the number of CEs detected by ZO1 labeling in Connexin mutants was indistinguishable from that observed in wildtype, while those detected by antibodies for the mutated Connexin were significantly reduced (*Figure 1—figure supplement 1I*). In addition, the zebrafish genome contains two additional homologous Connexin genes, *gjd2b/Cx35.1* and *gjd1b/Cx34.7*. We found that animals that were homozygous mutant for these two genes had normal Connexin and ZO1 labeling at CEs (*Figure 2—figure supplement 1G–L*). Similarly, we previously found there was no effect on M/CoLo synapses in the *gjd2b/Cx35.1* and *gjd1b/Cx34.7* mutants (*Miller et al., 2017*). Together, these results support a hierarchical relationship in the formation of neuronal GJs, in which ZO1 localizes to electrical synaptic contact sites where it is essential to robustly localize neuronal Connexins.

## ZO1b is required for electrical synaptic transmission

We next sought to examine the functional consequences of ZO1 and Connexin mutants, so we explored the properties of synaptic transmission at CEs during whole-cell recordings of the Mauthner cell. In wildtype zebrafish, extracellular stimulation of CE afferents near the posterior macula where they contact hair cells (*Figure 3A*) evoked a mixed synaptic response in the Mauthner cell composed of an early and large GJ-mediated electrical component followed by a delayed and smaller glutamatergic chemical response (*Figure 3B*; *Yao et al., 2014*). We first aimed to establish the functional consequences of removing the Connexins on this mixed synaptic response. Consistent with the presence of Cx35.5 and Cx34.1 at CEs, synaptic responses in *gjd2a/Cx35.5*$^{-/-}$ and *gjd1a/Cx34.1*$^{-/-}$ mutant zebrafish lacked a detectable electrical component, while exhibiting a response with the same delay as the chemical component of the mixed synaptic potential of wildtypes (*Figure 3C–E*). By contrast, electrical transmission was unaffected in *gjd2b/Cx35.1*$^{-/-}$ and *gjd1b/Cx34.7*$^{-/-}$ mutant zebrafish (*Figure 3E*; *Figure 3—figure supplement 1A-B*), as expected given our immunolabeling results (*Figure 2—figure supplement 1G-L*). The apparent chemical response in *gjd2a/Cx35.5*$^{-/-}$ and *gjd1a/Cx34.1*$^{-/-}$ mutants was blocked by extracellular application of a combination of the AMPA and NMDA glutamate receptor (GluR) antagonists cyanquixaline (CNQX) and D-2-Amino-5-phosphonovaleric acid (DAP5)(gray traces in *Figure 3C,D*; *Figure 3—figure supplement 1C*). No change in membrane potential was observed after application of the blockers. To confirm

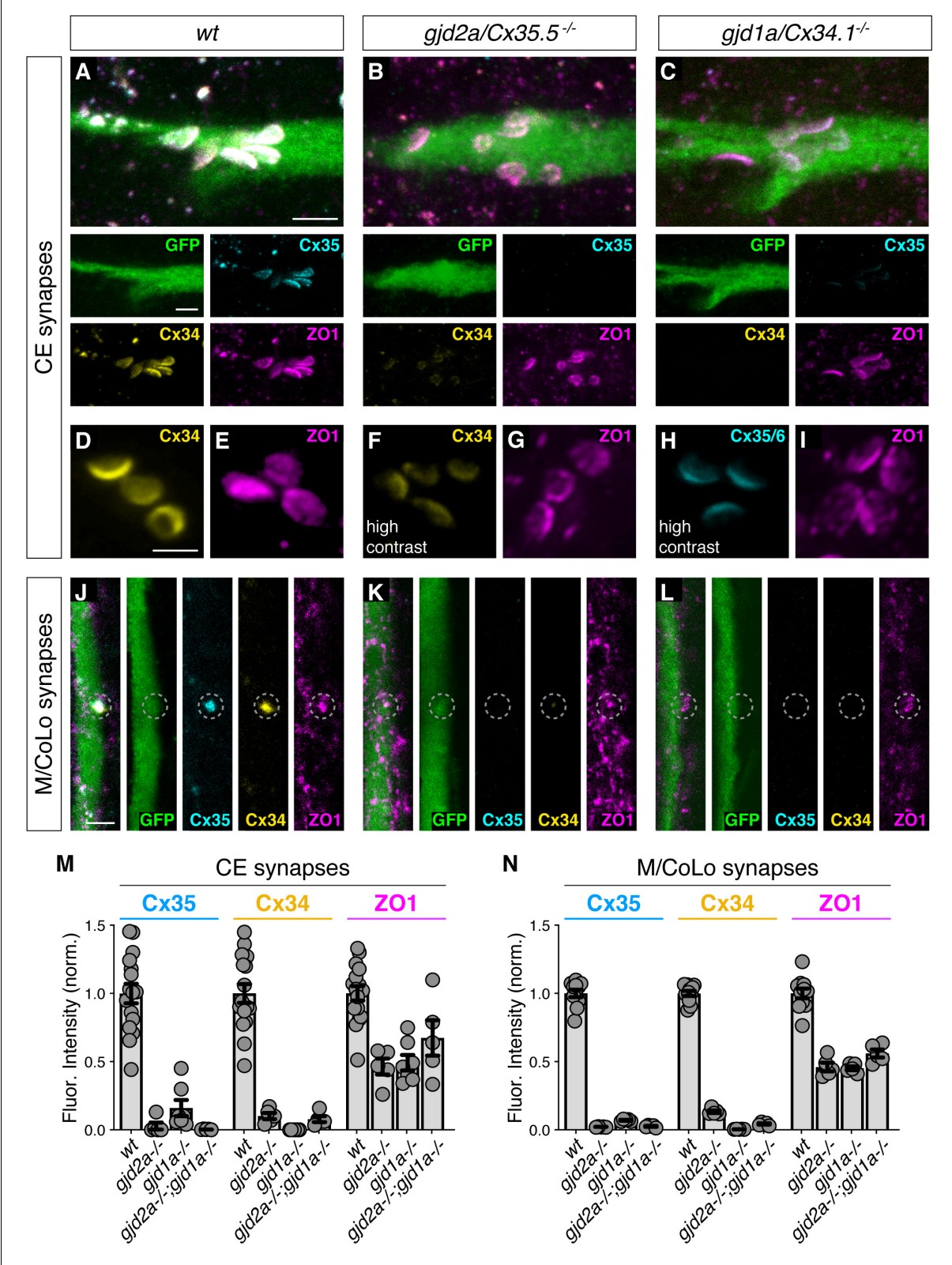

**Figure 2.** Localizing ZO1b to electrical synapses occurs independent of Connexins. (A–L) Confocal images of Mauthner circuit neurons and stereotypical electrical synaptic contacts in 5-day-post-fertilization, *zf206Et* zebrafish larvae from *wt* (A,D,E,J), *gjd2a/Cx35.5⁻ᐟ⁻* mutant (B,F,G,K), and *gjd1a/Cx34.1⁻ᐟ⁻* mutant animals (C,H,I,L). In panels (A–C,J–L) animals are stained with anti-GFP (green), anti-zebrafish-Cx35.5 (cyan), anti-zebrafish-Cx34.1 (yellow), and anti-human-ZO1 (magenta). In panels (D–I) animals are stained individually with the indicated antibody and in (F,H) the contrast is

*Figure 2 continued on next page*

*Figure 2 continued*

increased. Scale bar = 2 μm in all images. (A–C) Images of the stereotypical location of CE contact sites on the Mauthner lateral dendrite. Images are maximum-intensity projections of ~5 μm and neighboring panels show individual channels. (D–I) Images of the Mauthner CEs stained for individual electrical synapse components. Images are maximum-intensity projections of ~D ~ 2.66 μm, E ~ 1.90 μm, F ~ 1.90 μm, G ~ 0.72 μm, H ~ 2.28 μm, I ~ 2.16 μm. (F,H) Increased contrast for the Connexin channel reveals the residual staining at the synapses. (J–L) Images of the sites of contact of Mauthner/CoLo processes in the spinal cord. Images are individual Z-sections. Neighboring panels show individual channels. (M,N) Quantification of Cx35.5 (cyan), Cx34.1 (yellow), and ZO1 (magenta) fluorescence intensities at CE (M) and M/CoLo (N) synapses for the noted genotypes. *wt* data has been combined from all experiments. Individual data can be found in the *Figure 2—source data 1*. The height of the bar represents the mean of the sampled data normalized to the *wt* average. Circles represent the normalized value of each individual animal (CE synapse *wt, mut* paired experiments: *wt* n = 5, *gjd2a*[-/-] n = 5, *wt* n = 4, *gjd1a*[-/-] n = 7, *wt* n = 7, *gjd2a*[-/-]; *gjd1a*[-/-] n = 5; M/CoLo synapse *wt, mut* paired experiments: *wt* n = 5, *gjd2a*[-/-] n = 5, *wt* n = 3, *gjd1a*[-/-] n = 5, *wt* n = 3, *gjd2a*[-/-]; *gjd1a*[-/-] n = 5). Error bars are ± SEM. For each comparison, *wt* and mutant values are significantly different (Welch's t-test, p<0.01), except for the *wt* to *gjd2a*[-/-]; *gjd1a*[-/-] (Cx35.5/Cx34.1) double mutant comparison for ZO1 staining at CEs (p=0.842). Associated experimental statistics can be found in *Figure 2—source data 1*.

The online version of this article includes the following source data and figure supplement(s) for figure 2:

**Source data 1.** Source data for *Figure 2*.
**Figure supplement 1.** Characterization of Connexin mutants.
**Figure supplement 1—source data 1.** Source data for *Figure 2—figure supplement 1*.

that the chemical synaptic potential observed in *gjd2a/Cx35.5*[-/-] and *gjd1a/Cx34.1*[-/-] mutants arises from the stimulation of CEs lacking electrical transmission, we examined if blocking GJs with meclofenamic acid (MA) could reproduce the observed synaptic phenotype. We found that application of MA to wildtype recapitulated the *gjd2a/Cx35.5*[-/-] and *gjd1a/Cx34.1*[-/-] mutant phenotype, as the characteristic mixed synaptic response was replaced by a larger chemical synaptic response that was sensitive to GluR antagonists (*Figure 3F*; *Figure 3—figure supplement 1C*). We also observed a small hyperpolarization of the Mauthner cell (from −80.2 ± 0.7 mV in control to −84 ± 1 mV in MA; p=0.04, n = 5), likely resulting from the action of MA on other membrane channels. We conclude that together Cx35.5 and Cx34.1 generate functional neuronal GJs at CEs.

We next examined the properties of synaptic transmission in ZO1 mutants. Strikingly, and consistent with the requirement for Connexin localization at contact sites (*Figure 1*), *tjp1b/ZO1b*[-/-] mutants exhibited a failure in electrical transmission. The synaptic phenotype was indistinguishable from *gjd2a/Cx35.5*[-/-] and *gjd1a/Cx34.1*[-/-] mutants, and similarly consisted of a single, delayed response that was blocked by GluRs antagonists (*Figure 3G*; *Figure 3—figure supplement 1D*). This functional deficit was specific for ZO1b, as *tjp1a/ZO1a*[-/-] mutant fish exhibited normal mixed synaptic responses (*Figure 3H,I*). These findings indicate the specificity of the mutants to electrical synapses as glutamatergic transmission at CEs remained intact. Accordingly, neurotransmitter release properties (*Figure 3—figure supplement 1E*) and the localization of GluR2/3 receptors at these terminals were not affected in mutant fish (*Figure 3—figure supplement 2*). We conclude that the presence of the ZO1b scaffold protein is critical for electrical transmission at CEs, even though its absence does not prevent the formation of glutamatergic synapses coexisting within the same contact.

To investigate the extent of the deficit on electrical transmission within the brainstem network, we investigated whether Connexin and ZO1 mutations affected other synaptic contacts onto the Mauthner cell. The Mauthner cell receives mixed synaptic inputs from a variety of descending and ascending sensory modalities, including visual information from the optic tectum and somatic information from the spinal cord (*Dunn et al., 2016*; *Faber and Pereda, 2011*; *Kimmel et al., 1981*; *Korn and Faber, 2005*). For this purpose, we recorded spontaneous synaptic responses that represent the activity of most, if not all, synaptic inputs received by this cell. Electrically and chemically mediated spontaneous responses are easily differentiated due to their dramatically different duration (*Figure 4A*). Thus, for automated detection purposes, we defined fast spontaneous events (<1.1 ms), which represent the electrical coupling of presynaptic spikes, as electrical responses and slow spontaneous events (>1.1 ms) as chemical responses. Fast events were nearly absent in *tjp1b/ZO1b*[-/-], *gjd2a/Cx35.5*[-/-], and *gjd1a/Cx34.1*[-/-] mutants (*Figure 4B,C*), and were dramatically reduced in wild-type fish by application of MA (*Figure 4C*), confirming that they represent electrical transmission. Slow events that remained after losing the electrical component were dramatically reduced by application of GluRs antagonists (*Figure 4D*), confirming that they represent chemical responses.

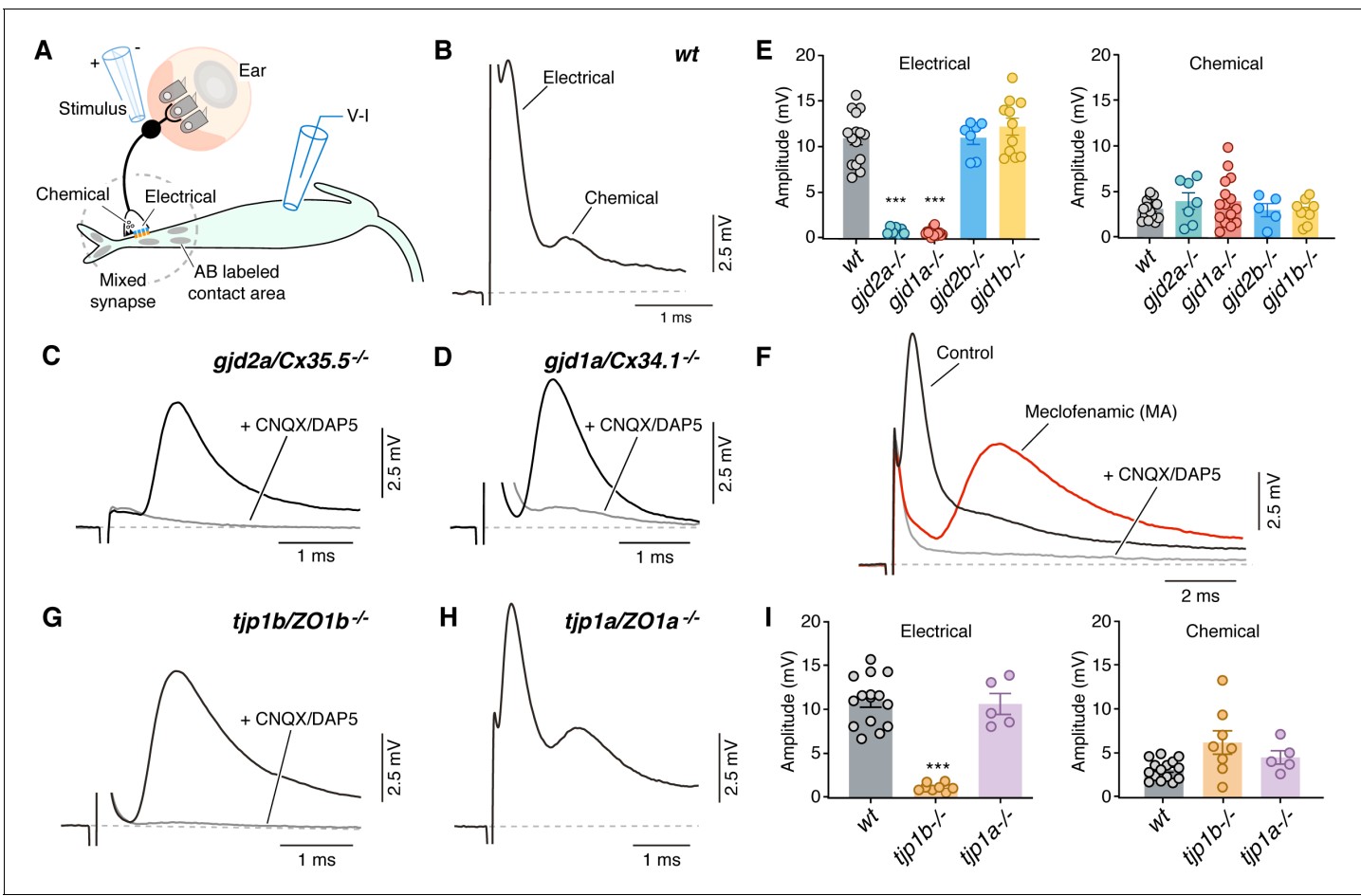

**Figure 3.** Electrical synaptic transmission at CEs requires ZO1b. (A) Diagram illustrates the experimental paradigm to examine synaptic transmission. (B) The 'mixed' synaptic response in the Mauthner cell evoked by extracellular stimulation of auditory afferents known as club endings (CEs) is composed in *wt* zebrafish larvae of an early electrical and a delayed chemically mediated response (membrane potential = −79 mv). Traces here and elsewhere represent the average of at least 10 single synaptic responses. (C,D) *gjd2a/Cx35.5⁻/⁻* and *gjd1a/Cx34.1⁻/⁻* mutant zebrafish had no detectable electrical component (black traces). The remaining synaptic response was blocked by bath application of CNQX and DAP5 (20 µM each) that block AMPA and NMDA glutamate receptors, respectively (membrane potential = −83.2 and −81 mv, respectively). (E) Bar graphs summarize the maximal amplitude (mean ± SEM), at a stimulation strength at which all CEs are activated, for the electrical and chemical components in *wt* and Connexin mutant zebrafish. Left, electrical: *wt*: 10.9 ± 0.7 mV (n = 15); *gjd2a/Cx35.5⁻/⁻*: 0.8 ± 0.1 mV (p<0.0001, n = 7); *gjd1a/Cx34.1⁻/⁻*: 0.6 ± 0.1 mV (p<0.00001, n = 15); *gjd2b/Cx35.1⁻/⁻*: 11.0 ± 0.7 (n = 7); *gjd1b/Cx34.7⁻/⁻*: 12.2 ± 0.9 mV (n = 11). The values in mutants lacking electrical transmission represent the membrane potential measured at the delay, which show the expected electrical component. Right, chemical: *wt*: 3.1 ± 0.3 mV (n = 15); *gjd2a/Cx35.5⁻/⁻*: 3.9 ± 0.9 mV (n = 7); *gjd1a/Cx34.1⁻/⁻*: 3.9 ± 0.7 mV (n = 15); *gjd2b/Cx35.1⁻/⁻*: 3.0 ± 0.7 mV (n = 5); *gjd1b/Cx34.7⁻/⁻*:2.9 ± 0.4 mV (n = 9). (F) Blocking electrical transmission recapitulates *gjd2a/Cx35.5⁻/⁻* and *gjd1a/Cx34.1⁻/⁻* synaptic phenotypes. Synaptic responses are superimposed and obtained before (black trace) and after (red trace) adding Meclofenamic acid (MA, 200 µM) to the perfusion solution. The remaining synaptic response was blocked (gray trace) after adding CNQX/DAP5 (20 µM each) to the perfusion solution (membrane potential = −81 mv). (G) *tjp1b/ZO1b⁻/⁻* zebrafish lack electrical transmission (black trace). The remaining synaptic potential was blocked by CNQX/DAP5 (20 µM each; gray trace) (membrane potential = −82 mv). (H) Synaptic responses in *tjp1a/ZO1a⁻/⁻* zebrafish show both electrical and chemical components (membrane potential = −87 mv). (I) Bar graphs illustrate the maximal amplitude (mean ± SEM) for the electrical and chemical components of the synaptic response in wt and ZO1 mutant zebrafish. Left, Electrical: *tjp1b/ZO1b⁻/⁻*: 1.1 ± 0.2 mV (p-value<0.0005, n = 8); *tjp1a/ZO1a⁻/⁻*: 10.6 ± 1.2 mV (n = 5). Right, chemical: *tjp1b/ZO1b⁻/⁻*: 6.2 ± 1.3 mV (n = 8); *tjp1a/ZO1a⁻/⁻*: 4.5 ± 0.8 mV (n = 5). Associated experimental statistics can be found in *Figure 3—source data 1*.

The online version of this article includes the following source data and figure supplement(s) for figure 3:

**Source data 1.** Source data for *Figure 3*.
**Figure supplement 1.** Electrophysiological characterization of Connexin and ZO mutants.
**Figure supplement 1—source data 1.** Source data for *Figure 3—figure supplement 1*.
**Figure supplement 2.** GluR2/3 localization is unaffected in ZO and Connexin mutants.

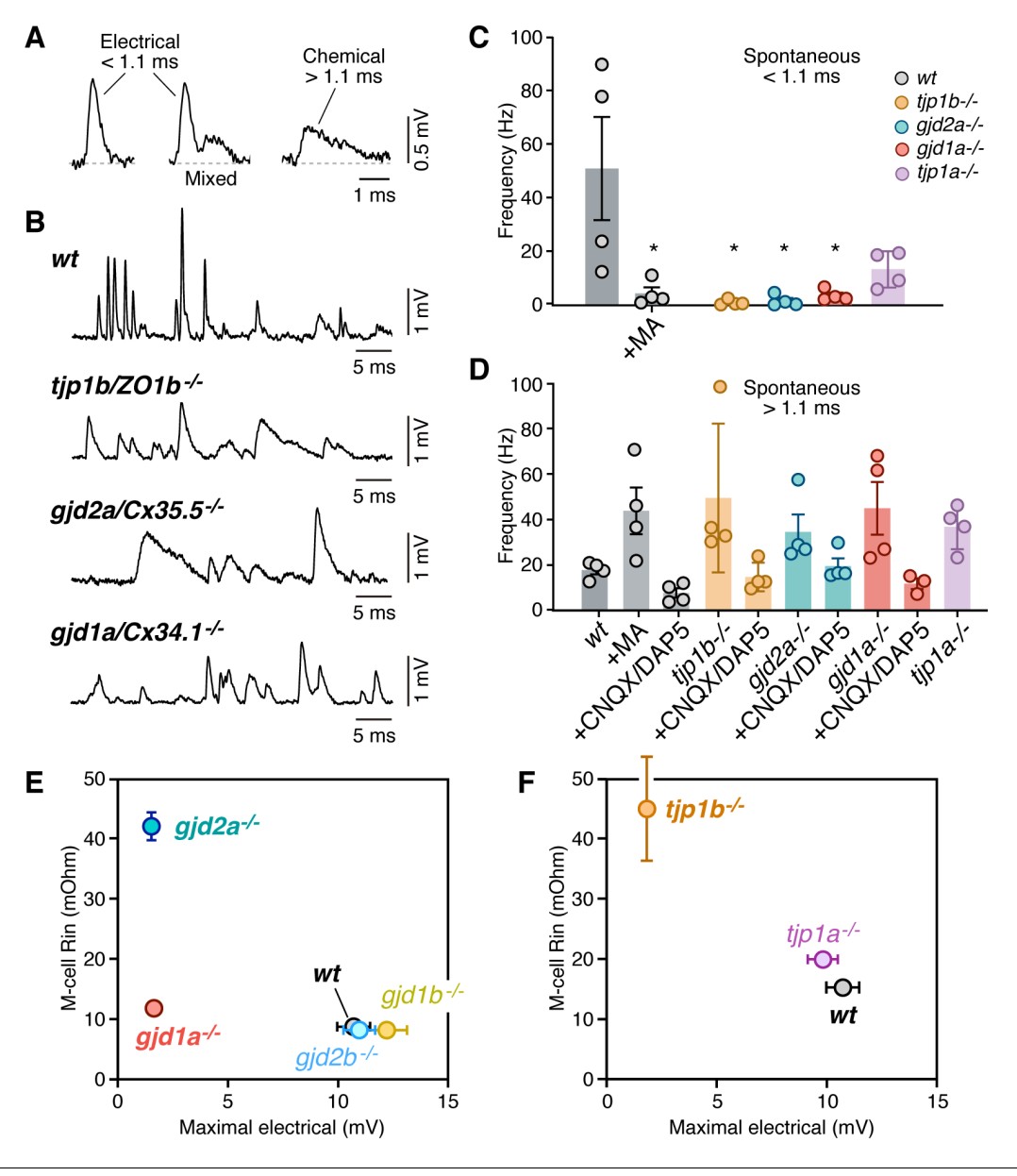

**Figure 4.** Lack of electrical transmission in *tjp1b/ZO1b*−/− is widespread and alters M-cell excitability. (**A**) Spontaneous responses in wildtype (wt) zebrafish can be electrical, chemical, or mixed. Spontaneous electrical and chemical responses were identified for automated detection by their duration: electrical responses were brief (<1.1 ms), whereas chemical responses were longer lasting (>1.1 ms). Mixed responses combined both. (**B**) Representative single traces of spontaneous synaptic activity obtained from the Mauthner cells of *wt*, *tjp1b/ZO1b*−/−, *gjd2a/Cx35.5*−/−, and *gjd1a/Cx34.1*−/−. Note the lack of short-lasting spontaneous responses in mutant zebrafish (membrane potential = −89, –87, and −89 mV, respectively). (**C**) Bar graph summarize the frequency in Hz (mean ± SEM; each n represents a fish) of the spontaneous short-lasting (<1.1 ms) electrical responses in *wt*, *tjp1b/ZO1b*−/−, *gjd2a/Cx35.5*−/−, *gjd1a/Cx34.1*−/− and *tjp1a/ZO1a*−/− zebrafish. The frequency of events in *wt* zebrafish was 50.9 ± 19.2 Hz (n = 4) and was reduced by MA (200 µM) to 4.2 ± 2.3 Hz (p<0.05). The variability between WT fish reflects different states of the network. The frequency was dramatically reduced in mutant zebrafish lacking electrical transmission: *tjp1b/ZO1b*−/−: 0.85 ± 0.5 Hz (n = 4; p<0.05); *gjd2a/Cx35.5*−/−: 1.5 ± 1.0 Hz (n = 4; p<0.05); *gjd1a/Cx34.1*−/−: 3.4 ± 1.1 Hz (n = 4, p<0.05). Although reduced, the change was not significant in *tjp1a/ZO1a*−/−: 13.2 ± 3.4 Hz (n = 4). (**D**) Long-lasting (>1.1 ms) chemical responses. The frequency of events in *wt* zebrafish was 17.8 ± 1.8 Hz (n = 4) and increased after MA to 43.8 ± 10.2 Hz (n = 4; p<0.05). The frequency was also increased in mutant zebrafish: *tjp1b/ZO1b*−/−: 49.5 ± 16.3 Hz (n = 4, p<0.05); *gjd2a/Cx35.5*−/−: 34.6 ± 7.7 Hz (n = 4, p<0.05);

*Figure 4 continued*

*gjd1a/Cx34.1*[-/-]: 45.0 ± 11.6 Hz (n = 4, p<0.05); *tjp1a/ZO1a*[-/-]: 36.8 ± 4.9 Hz (n = 4, p<0.05). Spontaneous events > 1.1 ms were greatly reduced by glutamate receptor antagonists (20 μM, CNQX/DAP5). The remaining responses likely represent depolarizing inhibitory responses prominent in the Mauthner cell. (E–F) Changes in the excitability of the Mauthner cell in Connexin and ZO1 zebrafish mutants. The graphs plot the input resistance of the Mauthner cell ($R$–$in$) vs the maximal amplitude of the electrical component of the synaptic response for *wt* and Connexin (E) and ZO1 (F) mutants. See *Table 1* for values of Rn and *Figure 3* for those of maximal electrical amplitude. Bars represent standard deviation. Associated experimental statistics can be found in for *Figure 4— source data 1*.

The online version of this article includes the following source data for figure 4:

**Source data 1.** Source data for *Figure 4*.

We conclude that the deficit in electrical transmission observed in mutant zebrafish is likely to be widespread within hindbrain and spinal cord circuits.

Given the primary role of the Mauthner cell in triggering escape responses, we also investigated possible changes in cellular excitability in mutant fish. Interestingly, the frequency of slow, chemical responses was increased in mutants and MA-treated animals (*Figure 4D*), even though we observed no changes on presynaptic neurotransmitter release properties (*Figure 3—figure supplement 1E*). This suggests that the lack of electrical transmission could have enhanced the detection of smaller amplitude chemical responses. We posit this could arise from electrical coupling influencing neuronal excitability, as the loss of neuronal GJs in mutants would increase the input resistance of the Mauthner cell (*Alcamí and Pereda, 2019*). Thus, during whole cell recordings, we determined the input resistance ($R_{in}$), rheobase, resting potential ($V_{rest}$), and firing threshold ($V_{threshold}$) of the Mauthner cells from wildtype and mutant zebrafish (*Table 1*). The input resistance, the main determinant of neuronal excitability, was increased in *tjp1b/ZO1b*[-/-], *gjd2a/Cx35.5*[-/-], and *gjd1a/Cx34.1*[-/-] mutants. Accordingly, the rheobase, a parameter negatively correlated with neuronal excitability, was decreased in these mutant animals (*Table 1*). We found that the change in $R_{in}$ correlated with the lack of electrical transmission and was not observed in fish that retained electrical transmission (*Figure 4E,F*; note however that $R_{in}$ in *tjp1a/ZO1a*[-/-] was found to slightly increase). Differences in magnitude of the effects observed on $R_{in}$ could be due to distinct compensatory mechanisms in each case or to the mutations affecting channels contributing to leak conductance in the Mauthner cell. Thus, the lack of electrical transmission in *tjp1b/ZO1b*[-/-] fish rendered synaptic transmission exclusively mediated by a relatively delayed glutamatergic response and more excitable Mauthner cells. Both deficits likely influence the behavioral responses generated by the Mauthner cell and its associated network.

**Table 1.** Mauthner cell electrophysiological properties from wildtype (wt), Connexin, and ZO-1 mutant zebrafish, Related to *Figure 4*.

Average measurements of resting potential ($V_{rest}$), firing threshold ($V_{threshold}$), input resistance ($R_{in}$) and Rheobase obtained in Mauthner cells of *wt*, *tjp1b/ZO1b*[-/-], *tjp1a/ZO1a*[-/-], *gjd2a/Cx35.5*[-/-], *gjd1a/Cx34.1*[-/-], *gjd2b/Cx34.7*[-/-], and *gjd1b/Cx35.1*[-/-] zebrafish. Each 'n' represents a fish (only one Mauthner cell was recorded in each fish). Associated experimental statistics can be found in *Table 1—source data 1*.

| Ephys. Prop. | *wt* (n = 10) | *tjp1b*[-/-] (n = 8) | *tjp1a*[-/-] (n = 5) | *gjd2a*[-/-] (n = 6) | *gjd1a*[-/-] (n = 9) | *gjd2b*[-/-] (n = 5) | *gjd1b*[-/-] (n = 6) |
|---|---|---|---|---|---|---|---|
| $V_{rest}$ (mV) | −83.5 ± 1.6 | −73.7 ± 2.2 | −85.8 ± 1.8 | −89.3 ± 2.4 | −86.9 ± 1.3 | −83.9 ± 0.9 | −84.6 ± 1.5 |
| $V_{threshold}$ (mV) | −59.1 ± 2.3 | −46.5 ± 1.7 | −55.3 ± 3.3 | −61.8 ± 2.2 | −60.5 ± 1.3 | −56.5 ± 3.6 | −58.3 ± 2.9 |
| Rin (MOhm) | 6.1 ± 0.8 | 45.4 ± 8.6 | 11.7 ± 1.3 | 42.0 ± 2.3 | 11.8 ± 0.9 | 5.5 ± 0.6 | 5.5 ± 0.7 |
| Rheobase (nA) | 4.0 ± 0.4 | 1.5 ± 0.2 | 2.7 ± 0.4 | 1.2 ± 0.2 | 2.6 ± 0.2 | 4.4 ± 0.1 | 4.9 ± 0.6 |

The online version of this article includes the following source data for  Table 1:
**Source data 1.** Source data for *Table 1*.

## ZO1b is essential for appropriate Mauthner-cell-initiated startle responses

Since *tjp1b/ZO1b*[-/-] mutants had functional deficits in electrical transmission, but not chemical, we assessed the consequences on the behavioral output of the Mauthner cell network. Animals were placed into individual chambers of a multi-well testing stage and presented with acoustic-vibrational stimuli to elicit Mauthner-dependent startle responses (*Wolman et al., 2015*). Movements were captured with a high-speed camera (1000 frames per second) and analyzed with FLOTE software to automatically track and measure the kinematics (body movements) of responses (*Burgess and Granato, 2007*). In this paradigm, wild-type fish exhibit two types of escape responses: (1) Mauthner-cell-dependent short-latency C-bends (SLCs, hereafter referred to as 'startles') and (2) Mauthner-cell-independent long-latency C-bends (LLCs). These two behavioral responses were automatically distinguished using well-established kinematic parameters (*Burgess and Granato, 2007*). Larvae generated from crossing *tjp1b/ZO1b*[+/-] heterozygous animals were tested and analyzed blind to genotype with subsequent post-hoc identification. We found that *tjp1b/ZO1b*[-/-] mutants startled to strong acoustic stimuli (25.9 dB) as frequently as their wildtype siblings (*Figure 5A*). While these turns were classified as startles, we found that the *tjp1b/ZO1b*[-/-] mutants initiated their responses ∼ 2 ms slower than their wildtype siblings (*Figure 5B*). A delayed behavioral response in consistent with our electrophysiological findings indicating that the mutant escape network operates with longer synaptic delays due to the lack of electrical transmission (*Figure 3*). In mutant animals, we found that the escapes were often performed with the normal startle kinematic parameters, particularly the maximum angle of the turn and the maximum angular velocity of the response, albeit occurring later than in wild-type siblings (*Figure 5C–H*). However, we found a subset of mutant responses (∼15%) that showed abnormally shallow and slow turns (*Figure 5C,D* red arrows). These abnormal responses suggested deficits in performing the stereotyped C-bend elicited by the Mauthner-cell network, and so we reanalyzed the video data from these responses. In these startles, we observed that the mutants displayed abnormal postures where the body would bend slightly to one side, creating 'kinked' or 'S-shaped' postures (*Figure 5I–L*). We note that the phenotypes observed in the *tjp1b/ZO1b*[-/-] mutants are strikingly similar to those we previously observed in *gjd2a/Cx35.5*[-/-] and *gjd1a/Cx34.1*[-/-] mutants (*Miller et al., 2017*). Such kinked body shapes are reminiscent of startles following CoLo neuron ablation (*Satou et al., 2009*). Based on these data, we conclude that electrical synapses are essential for generating the speed and coordination of the Mauthner-induced startle response.

The electrophysiological analysis of *tjp1b/ZO1b*[-/-] mutants also revealed that Mauthner cells showed increased excitability (*Figure 4*), suggesting animals may be hypersensitive with a lower threshold of response for environmental stimuli. To examine this possibility, we presented larvae with 60 pseudo-randomized stimuli, 10 at six different intensities with a 20 s inter-stimulus interval to eliminate habituation (*Wolman et al., 2015*). We then assessed the frequency with which animals responded to the stimuli with turns (*Figure 5M–O*). As stimulus intensity increased, both wildtype siblings and *tjp1b/ZO1b*[-/-] mutants increased the likelihood of performing a startle response. However, across the mid-range of stimulus intensities, *tjp1b/ZO1b*[-/-] mutants were more likely to respond with a startle than their wild-type siblings (*Figure 5M,O*). The increased tendency of mutants to perform Mauthner-dependent startles came at the expense of Mauthner-independent LLC responses, which were nearly absent in mutants (*Figure 5N*). We conclude that electrical synapses alter the sensitivity of Mauthner cells to environmental stimuli and contribute to the innate startle threshold, which alters the probability of eliciting a startle or LLC response. These results lend additional evidence for the critical role of ZO1b for creating functional electrical synapses, ultimately contributing to appropriately balanced neural network function and behavior.

## ZO1b interacts exclusively with Cx34.1 in vivo

Mutant zebrafish revealed a hierarchical relationship between ZO1b and neuronal Connexins (*Figures 1* and *2*), so we next investigated the mechanisms underlying this relationship. ZO1 is a membrane-associated guanylate kinase (MAGUK) scaffold protein and contains PSD95/Dlg/ZO1 (PDZ) protein-protein interaction domains that bind to PDZ binding motifs (PBMs) (*Zhu et al., 2016*). Previous work demonstrated that the C-terminal four amino acids of mouse Cx36 and perch Cx35 compose PBMs that are essential for interacting with ZO1 (*Flores et al., 2008*; *Li et al., 2004*). Given

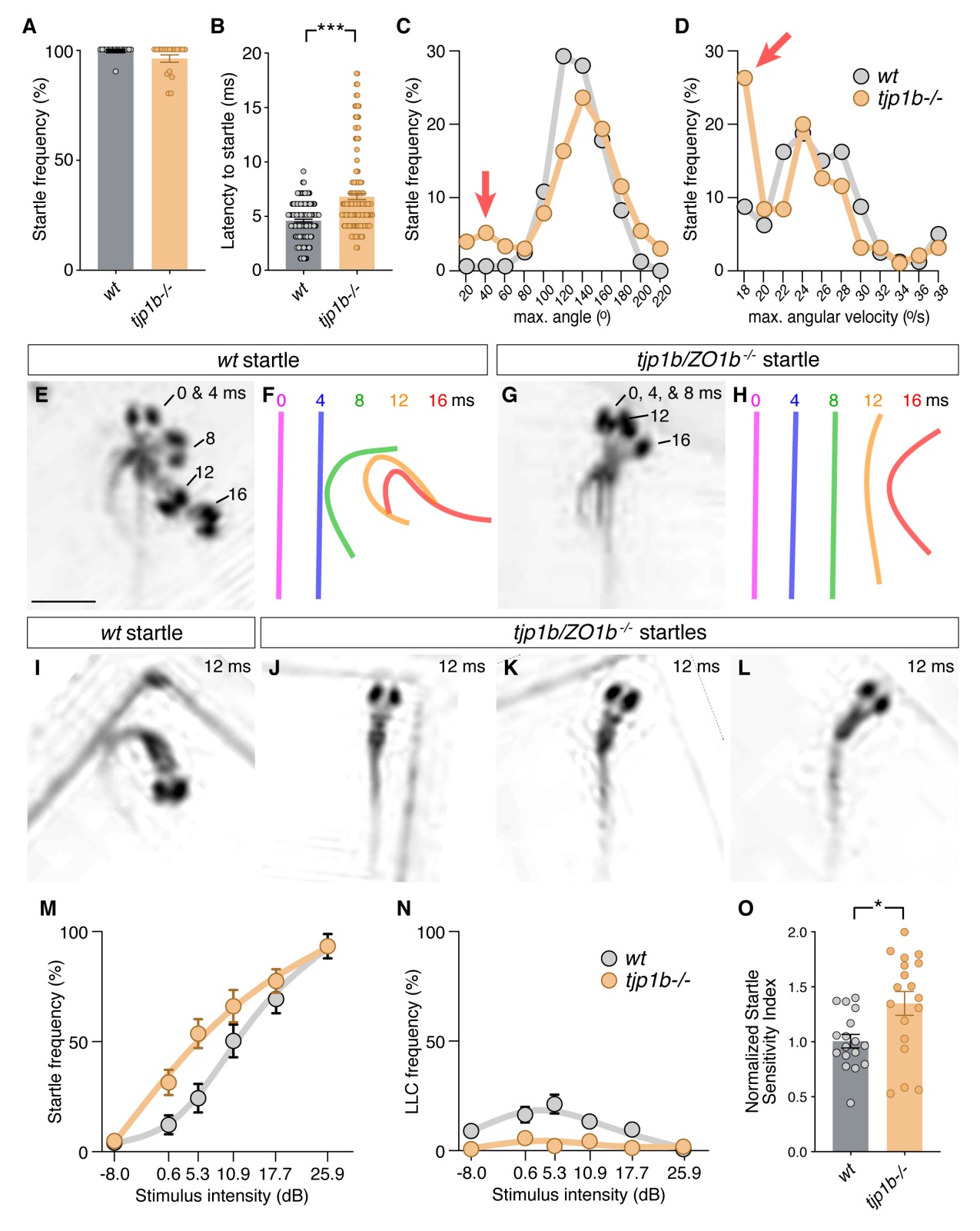

**Figure 5.** Mauthner-cell-initiated escape response parameters require ZO1b. (**A**) Frequency of responses classified as Mauthner-initiated startles in *wt* and *tjp1b/ZO1b⁻/⁻* mutants. Bar graphs show mean ± SEM. Each circle represents an individual animal's average frequency of responses to 10 independent trials (*wt* n = 17, *tjp1b/ZO1b⁻/⁻* n = 18; Mann-Whitney test p=0.0947). (**B**) Latency of initiating startles in all individual trials. Bar graphs represent data as mean ± SEM with each circle representing individual latencies (*wt* n = 157, *tjp1b/ZO1b⁻/⁻* n = 165; Mann-Whitney test p<0.0001). (**C,D**)

*Figure 5 continued on next page*

Figure 5 continued

Kinematic analysis of the maximum turn angle (C) and the maximum angular velocity (D) of the startles plotted as frequency of events within the indicated bin. Red arrows indicate abnormal shallow angle and low velocity turns exhibited by *tjp1b/ZO1b*$^{-/-}$ mutants. (E–H) Time-lapse of *wt* (E,F) and *tjp1b/ZO1b*$^{-/-}$ mutant (G,H) startles. Scale bar = 1 mm. Individual snapshots taken at the indicated times (ms = milliseconds) are overlaid on an individual image (E,G). A line representing the midline body axis at each time was drawn to indicate the movement (F,H). (I–L) A *wt* startle bend at its maximum angle (I) compared to abnormally shaped bends executed by *tjp1b/ZO1b*$^{-/-}$ mutant larvae (J–L). (M–N) Mauthner-induced startle frequency (M) and long-latency C-bend (LLC) frequency (N) for 10 trials at six intensities with fit curves for *wt* and *tjp1b/ZO1b*$^{-/-}$ mutants. Each symbol represents data as mean ± SEM (*wt* n = 17, *tjp1b/ZO1b*$^{-/-}$ n = 18). (O) The startle sensitivity index is determined as the area under the curves for each individual animal in (M). Bar graphs represent data as mean ± SEM with each circle representing individual sensitivity indices (*wt* n = 17, *tjp1b/ZO1b*$^{-/-}$ n = 18; Mann-Whitney test p=0.03). Associated experimental statistics can be found in *Figure 5—source data 1*.

The online version of this article includes the following source data for figure 5:

**Source data 1.** Source data for *Figure 5*.

that the PBM sequence is conserved in zebrafish Cx35.5 and Cx34.1 proteins (*Figure 6A*), we tested whether these Connexins could mediate binding to zebrafish ZO1b. We cloned full-length sequences of *tjp1b/ZO1b*, *gjd2a/Cx35.5*, and *gjd1a/Cx34.1* and used heterologous expression to test for interactions between the scaffold and Connexins. HEK293T cells were co-transfected with mVenus-ZO1b and full-length Cx35.5 or Cx34.1. Using western blot analysis with antibodies specific to each Connexin, we found that both Cx34.1 and Cx35.5 were individually detected in mVenus-ZO1b immune complexes (*Figure 6B*, lanes 1,3) compared to control immunoprecipitates (*Figure 6—figure supplement 1A*, lanes 1,2,5,6). We further found that removing the presumptive PBMs, by deleting the C-terminal four amino acids from both Connexins, resulted in a loss of co-purifying Connexins from mVenus-ZO1b immunoprecipitates (*Figure 6B*, lanes 2,4; *Figure 6—figure supplement 1A*, lanes 4,8). Control blots demonstrated the ability of the Connexin antibodies to equally recognize both full-length and ΔPBM versions of the proteins (*Figure 6B*; *Figure 6—figure supplement 1A*, bottom input panels). We conclude that zebrafish Cx35.5 and Cx34.1 can interact with ZO1b in a PBM-dependent manner.

The ZO1b scaffold has three PDZ domains that could mediate the interaction with neuronal Connexins (*Figure 6A*). Previous studies testing the three mammalian ZO1 PDZ domains demonstrated that Cx35/36 PBMs exclusively bound to ZO1-PDZ1, while PDZ2 and PDZ3 did not interact (*Flores et al., 2008*; *Li et al., 2004*). Given the high degree of conservation of the zebrafish ZO1 PDZ1 domain, including the amino acids of the putative PBM binding pocket (*Figure 6A*), we tested whether zebrafish ZO1b-PDZ1 could directly interact with zebrafish neuronal Connexins. To examine this question, we isolated the minimal domains of each zebrafish protein, produced them in bacteria, and performed in vitro binding studies. We found that purified ZO1b-PDZ1 could be pulled down with a GST-Cx34.1 or a GST-Cx35.5 C-terminal intracellular-tail (*Figure 6C*, lanes 2,4), but not with control GST protein (*Figure 6C*, lane1). Further, this interaction was significantly decreased when the predicted PBM in the Connexin tails were removed (*Figure 6C*, lanes 3,5). We next used an overlay assay to compare the ability of ZO1b-PDZ1 and ZO1b-PDZ2 to bind to immobilized GST-Connexin tails. Similar to the binding assays, significant amounts of ZO1b-PDZ1 bound to GST-Cx34.1 and GST-Cx35.5 tails in a PBM-dependent manner (*Figure 6—figure supplement 1B*, left panels). By contrast, little ZO1b-PDZ2 bound to the GST-Cx34.1 or GST-Cx35.5 tails, particularly when compared to the non-neuronal Cx43 C-terminal tail (*Figure 6—figure supplement 1B*, right panels), which is known to bind PDZ2 (*Flores et al., 2008*; *Li et al., 2004*). We conclude that ZO1b utilizes the PDZ1 domain to directly interact with the neuronal Connexin PBMs.

Since ZO1b and the neuronal Connexins colocalized at electrical synapses (*Figure 1*), we next investigated whether these proteins interacted in vivo. We utilized adult zebrafish brains that maintain widespread electrical synapses, including those in Mauthner cell (*Kimmel et al., 1981*), and provide an abundant source to derive ZO1b immunoprecipitates. Homogenates derived from wild-type fish brains were immunoprecipitated with anti-ZO1 and control antibodies (mIgG). Immunoprecipitates demonstrated that Cx34.1 copurified with ZO1, whereas Cx35.5 did not copurify with the anti-ZO1 antibody (*Figure 6D*, lanes 1,2). To confirm that copurification of Cx34.1 was dependent upon ZO1b, we replicated the experiment using homogenates from *tjp1b/ZO1b*$^{-/-}$ mutants and found that Cx34.1 was lost in these immmunocomplexes (*Figure 6D*, lane4). We conclude that ZO1 preferentially interacts with Cx34.1 in vivo. We observed multiple ZO1 bands upon Western analysis of the

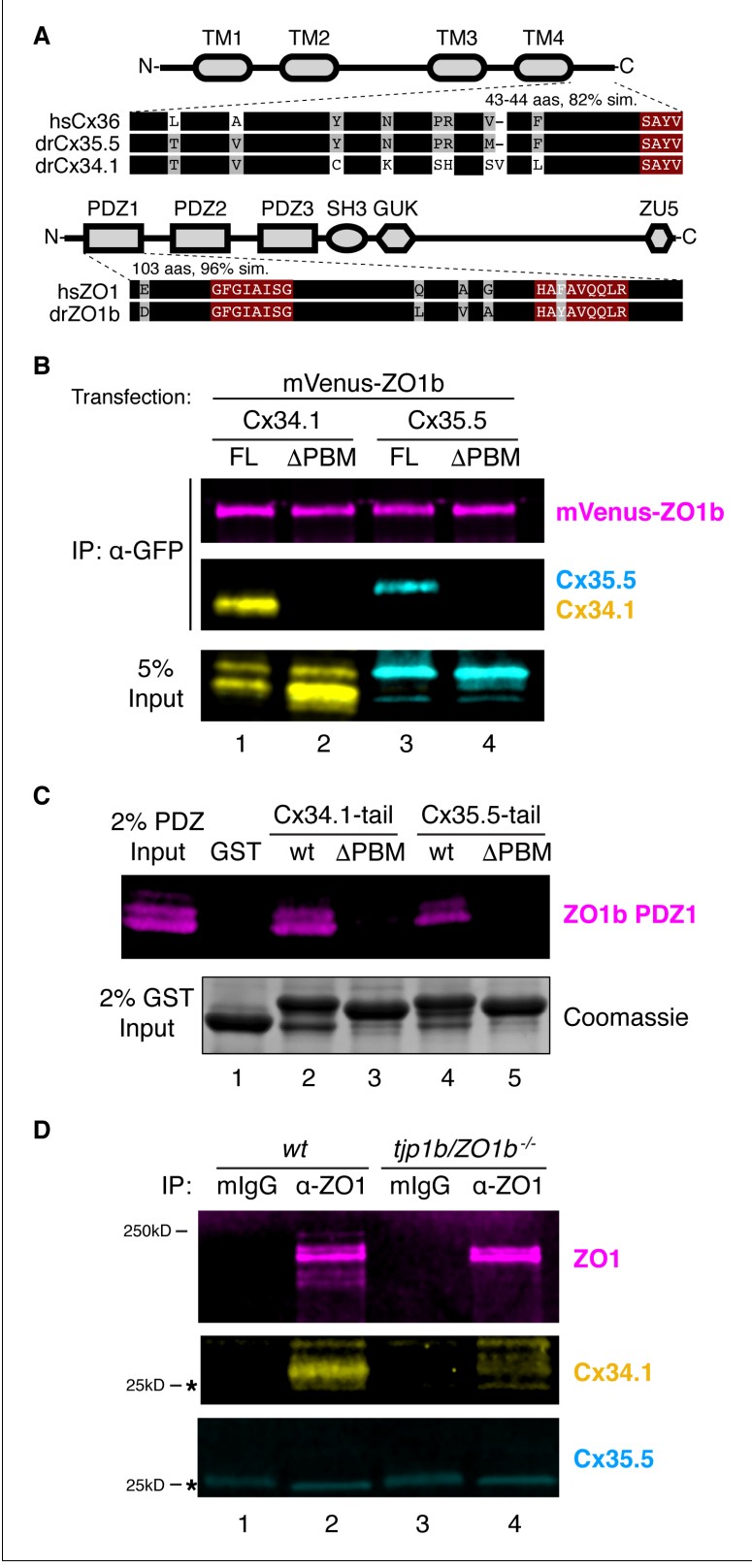

**Figure 6.** ZO1b scaffolds postsynaptic Cx34.1 in vivo. (**A**) Schematic, linear diagrams of Cx36 and ZO1 homologues. Domains are depicted as gray shapes; TM = transmembrane, PDZ, SH3, GUK, and ZU5 = protein-protein interaction modules; hs = *Homo sapiens*, dr = *Danio rerio*. Amino acid alignments are shown for the indicated expanded regions. Black bars represent conserved amino acids; non-conserved amino acids are
*Figure 6 continued on next page*

*Figure 6 continued*

indicated. Maroon boxed amino acids represent the conserved PDZ-binding motif (PBM) of Cx36-family proteins (top) or the predicted PDZ1 residues of the conserved ligand-binding cleft of ZO1-family proteins (bottom). (B) HEK293T/17 cells were transfected with plasmids to express mVenus-ZO1b and either full-length Cx34.1 (lane 1), Cx34.1-ΔPBM (lane 2), full-length Cx35.5 (lane 3), or Cx35.5-ΔPBM (lane 4). Lysates were immunoprecipitated with anti-GFP antibody and analyzed by immunoblot for the presence of mVenus-ZO1b using anti-GFP antibody (upper, magenta), Cx34.1 protein using Cx34.1-specific antibody (middle, yellow), or Cx35.5 protein using Cx35.5-specific antibody (middle, cyan). Total extracts (bottom, 5% input) were blotted for Connexin proteins to demonstrate equivalent expression and uniform antibody recognition of expressed proteins. Results are representative of three independent experiments. (C) Bacterially purified GST (lane 1), GST-Cx34.1-tail (lane 2), GST-Cx34.1-tail-ΔPBM (lane 3), GST-Cx35.5-tail (lane 4), or GST-Cx35.5-tail-ΔPBM (lane 5) was immobilized on glutathione beads and incubated with purified ZO1b PDZ1 domain. The tail regions used are depicted in the expanded regions in (A). Bound proteins were analyzed by immunoblot for the presence of ZO1b PDZ1 using anti-TEV cleavage site antibody (top, magenta). Equal loading of GST proteins is indicated by Coomassie staining (bottom, 2% input). Results are representative of three independent experiments. (D) Zebrafish brain extract from *wt* (lanes 1,2) or *tjp1b/ZO1b⁻/⁻* mutant (lanes 3,4) animals was immunoprecipitated with control whole mouse IgG (lanes 1,3) or anti-ZO1 antibody (lanes 2,4). Immunoprecipitates were analyzed by immunoblot for the presence of ZO1 using anti-ZO1 antibody (top, magenta), Cx34.1 using Cx34.1-specific antibody (middle, yellow), and Cx35.5 using Cx35.5-specific antibody (bottom, cyan). Asterisks (*) indicate antibody light chain. Results are representative of three independent experiments.

The online version of this article includes the following figure supplement(s) for figure 6:

**Figure supplement 1.** Biochemical characterization of ZO and Connexin interactions.

---

immunoprecipitates, several of which were lost in the *tjp1b/ZO1b⁻/⁻* mutants (**Figure 6D**, lanes 2,4). Since the ZO1 antibody was made against human protein, we reasoned it might be detecting the ZO1b protein and the highly similar ZO1a protein, so we sought evidence to confirm that ZO1b was the primary scaffold for Cx34.1 in vivo. Upon examining ZO1 immunocomplexes from *wildtype*, *tjp1b/ZO1b⁻/⁻*, and *tjp1a/ZO1a⁻/⁻* brains, we found that only ZO1b deficiency resulted in concomitant loss of Cx34.1 (**Figure 6—figure supplement 1C**). Taken together, we conclude that ZO1b preferentially interacts with Cx34.1 in vivo, despite the fact that the scaffold can interact with either neuronal Connexin.

## ZO1b localizes and functions postsynaptically at electrical synapses

Next, we determined the functional relevance for a preferential ZO1b/Cx34.1 interaction at zebrafish electrical synapses. We previously observed that Cx35.5 localizes presynaptically in axons, while Cx34.1 localizes postsynaptically in dendrites (*Miller et al., 2017*). Since ZO1b preferentially interacts with Cx34.1 in vivo (*Figure 6*), we speculated that ZO1b might share a similar dendritically compartmentalized localization. To directly examine this, we visualized the localization of ZO1b protein after inserting a V5 epitope at the N-terminus of the endogenous *tjp1b* locus (*Figure 7—figure supplement 1A*). We found that in transgenic *V5-tjp1b* larvae, V5 antibody staining colocalized with both Cx34.1 and Cx35.5 at CEs and M/CoLo synaptic contacts (*Figure 7—figure supplement 1B–E*). Moreover, we found no effect on the stereotyped patterns of Connexin staining at Mauthner electrical synaptic contact sites in homozygous *V5-tjp1b/V5-tjp1b* animals (*Figure 7—figure supplement 1D,E*), suggesting that V5-ZO1b is functional. While *V5-tjp1b* larvae permitted ZO1b visualization at electrical synapses, we could not discriminate whether it was localized asymmetrically at synaptic contacts due to the small size of these structures and the resolution limits of light microscopy. To overcome this, we utilized an alternate approach to address ZO1b compartmentalization, exploiting the fact that the Mauthner cell is both the postsynaptic partner at CEs and the presynaptic partner at the M/CoLo synapses (*Figure 7A*). We generated chimeric embryos by blastula transplantation (*Kemp et al., 2009*), extracted GFP and V5-ZO1b expressing transgenic cells from donor embryos, and transferred them to wild-type hosts (*Figure 7B*), allowing us to address V5-ZO1b localization within the Mauthner cell (*Figure 7C–E*). In animals containing only a donor-derived, V5-ZO1b-expressing Mauthner cell, we found that V5 staining was present at the CEs with Connexin staining, demonstrating that ZO1b was within the dendrite of the Mauthner cell and localized postsynaptically at electrical synapses (*Figure 7D*). Conversely, when we examined the M/CoLo synapses of these same embryos, we found that V5 staining was not present at these synaptic contacts

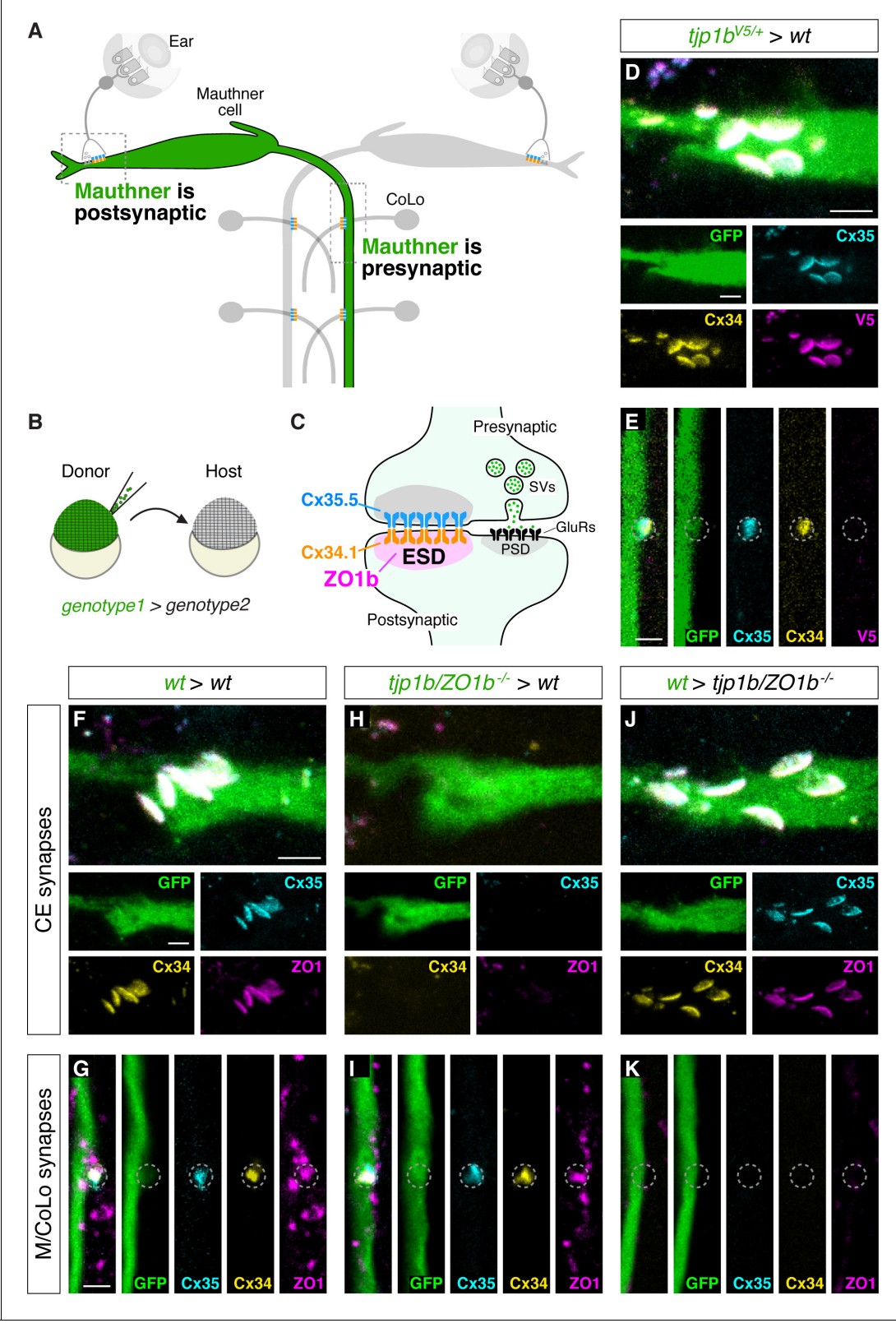

**Figure 7.** ZO1b localizes and functions postsynaptically. (**A**) Schematic of the Mauthner circuit in chimeric animals. One Mauthner cell is derived from the GFP-expressing donor (green), while other neurons derive from the non-transgenic host (gray). The image represents a dorsal view with anterior to the top. Electrical synapses denoted as yellow (Cx34.1) and cyan (Cx35.5) rectangles. Boxed regions indicate regions imaged for analysis. (**B**) Diagram of experiment in which GFP-expressing donor cells are transplanted into a non-transgenic host to create chimeric embryos. GFP-expressing cells are of

*Figure 7 continued on next page*

*Figure 7 continued*

*genotype1* while the rest of the cells in the chimeric embryo are derived from *genotype2*. (C) Diagram of a mixed electrical/chemical (glutamatergic) synapse summarizing data for ZO1b. ESD = electrical synapse density, see Discussion. (D–K) Confocal images of Mauthner circuit neurons and stereotypical electrical synaptic contacts in 5-day-post-fertilization, chimeric zebrafish larvae. Animals are stained with anti-GFP (green), anti-zebrafish-Cx35.5 (cyan), and anti-zebrafish-Cx34.1 (yellow). In panels (D–E), animals are stained with anti-V5 (magenta), and in (F–K) animals are stained with anti-human-ZO1 (magenta). The genotype of the donor cell (green, *genotype1*) and host (*genotype2*) varies and is noted above each set of images (*genotype1 > genotype2*). Images of CEs (D,F,H,J) are maximum-intensity projections of ~5 μm. Images of M/CoLo synapses (E,G,I,K) are single Z-slices. Neighboring panels show individual channels. Scale bar = 2 μm in all images.

The online version of this article includes the following source data and figure supplement(s) for figure 7:

**Figure supplement 1.** Characterization of ZO1b localization and function.

**Figure supplement 1—source data 1.** Source data for *Figure 7—figure supplement 1*.

despite the Mauthner cell expressing V5-ZO1b protein (*Figure 7E*). We note that in the non-chimeric, V5-ZO1b transgenic animals, V5 staining was observed at the M/CoLo synapses (*Figure 7—figure supplement 1C,E*), suggesting that ZO1b at these synapses derives from the postsynaptic CoLo. Our transplant experiments produce chimeric larvae in which only Mauthner, only CoLo, or both cells are derived from the transgenic donor embryos (*Figure 7—figure supplement 1F–H*). In larvae in which only CoLo expresses V5-ZO1b, we found that V5 staining is present at M/CoLo synapses (*Figure 7—figure supplement 1G*), confirming ZO1b's postsynaptic localization. We conclude that ZO1b is robustly compartmentalized within the somato-dendritic compartment of the neuron and asymmetrically localized on the postsynaptic side of the electrical synapse.

We then addressed whether ZO1b functions postsynaptically to facilitate Connexin localization at the electrical synapse. We utilized chimeric animals and again took advantage of Mauthner cell morphology. However, in these experiments, we transplanted GFP-expressing cells from *tjp1b/ZO1b^-/-* mutant donors into wildtype hosts, producing animals in which ZO1b was specifically removed from the Mauthner cell (*Figure 7F–I*). At CEs, the removal of ZO1b exclusively from the Mauthner cell resulted in the loss of staining for ZO1, Cx34.1, and Cx35.5 (*Figure 7H*; *Figure 7—figure supplement 1I*). The data indicates that ZO1b is required postsynaptically for the cell autonomous localization of Cx34.1 and non-autonomously for presynaptic Cx35.5 localization. By contrast, when we examined the same chimeric animals with ZO1b removed from the Mauthner cell but focused on the M/CoLo synapses, in which Mauthner is presynaptic, there was no effect on either Connexin or ZO1 staining (*Figure 7I*; *Figure 7—figure supplement 1J*), indicating that ZO1b is dispensable presynaptically. As further support for this compartmentalized scaffold function, we reasoned that resupplying ZO1b to the Mauthner cell in an otherwise *tjp1b/ZO1b^-/-* mutant animal would be sufficient to rescue Connexin localization at CEs, but not at M/CoLo synapses. To test this prediction, we transplanted GFP-expressing cells from wildtype donor embryos into *tjp1b/ZO1b^-/-* mutant hosts and identified animals with donor-derived Mauthner cells. In such chimeras, where the *tjp1b/ZO1b* gene is functional only in the Mauthner cell, CEs had normal staining for both postsynaptic ZO1 and Cx34.1 and also for presynaptic Cx35.5. By contrast, there was no rescue of staining for these proteins at M/CoLo synaptic contacts (*Figure 7J,K*; *Figure 7—figure supplement 1J*). We conclude that ZO1b is both necessary and sufficient postsynaptically for building the structure of the neuronal GJ channels. Taken together, we find that ZO1b is exclusively localized to the postsynaptic compartment where it functions both cell-autonomously and non-autonomously to localize Connexins and build functional neuronal gap junctions.

## Discussion

We exposed here a structural and regulatory scaffolding protein, ZO1b, that is essential for the formation and function of electrical synapses. Our data indicate an asymmetrical, compartmentalized, and hierarchical relationship between this scaffolding protein and the channel-forming Connexins. These findings challenge current perceptions of the functional and molecular organization of electrical synapses, calling for a new model that includes a primary role for the intracellular molecular scaffold in governing the formation of functional intercellular channels. Based on this evidence, we propose that electrical synapses likely constitute complex and asymmetric synaptic structures with features that parallel the molecular and functional organization of the PSD at chemical synapses.

## Hierarchical assembly of the electrical synapse

Despite the continuous nature of electrical transmission, the thousands of channels that make up GJ plaques found at electrical synapses are maintained by the active turnover of Connexin proteins (*Flores et al., 2012*), similar to neurotransmitter receptors at chemical synapses (*Carroll and Zukin, 2002*; *Chen et al., 2000*; *Ehlers, 2000*; *Lüscher et al., 1999*). Our findings show that ZO1b is required for the robust localization of Connexins, suggesting it functions to stabilize Connexin hemichannels at the synaptic site. Additionally, ZO1 can localize to electrical synapses in the absence of the Connexins, although its apparent concentration was diminished. This suggests that building robust neuronal GJ structures involves a reciprocal interaction between ZO1 and the Connexins. Strikingly, our results reveal that ZO1b is compartmentalized to the dendrite and functions asymmetrically at postsynaptic sites of the Mauthner cell circuit. This localization is consistent with ZO1b's preferential in vivo interaction with Cx34.1, as this Connexin is also localized and required postsynaptically at Mauthner cell electrical synapses (*Miller et al., 2017*). Despite the apparent autonomous ZO1b/Cx34.1 postsynaptic interaction at the GJ hemiplaque, *tjp1b/ZO1b-/-* mutants revealed that presynaptic Cx35.5 localization is also affected non-autonomously in the neighboring cell. This transsynaptic interaction likely occurs via the Connexins themselves, as mutations to the postsynaptic Cx34.1 prevented the robust localization of the presynaptic Cx35.5. Our results are complementary to recent analysis of the mouse rod/cone network, where removing Cx36 from one neuron of a coupled pair results in the failure of Connexin localization in the adjacent neuron (*Jin et al., 2020*). Taken together, our results reveal that ZO1b acts as a postsynaptic molecular scaffold that localizes Cx34.1 to the GJ hemiplaque, which in turn ensures Cx35.5 stabilization at presynaptic hemiplaques (*Figure 7C*). Whether presynaptic Cx35.5 lacks an in vivo scaffold, or utilizes another unidentified presynaptic scaffolding protein, remains unresolved. If the molecular function and organization of ZO1 revealed here applies to all electrical synapses, including those formed by homotypic channels between various homologous cellular processes (dendro-dendritic, somato-somatic, or axo-axonic), remains to be determined in future studies. Never-the-less, we posit that the molecular organization of the electrical synapse can be asymmetrically compartmentalized, thereby enabling preferential biochemical interactions at each side of the junction.

## ZO1's role in synaptic communication

Our results support the prediction that ZO1 likely plays distinct functional roles at GJs in different tissue types. ZO1 was first described at epithelial tight junctions and later shown to interact with various Connexins, notably with Cx43, a widespread Connexin expressed in many non-neuronal cell types (*Giepmans, 2004*). Evidence from cell expression systems suggested that ZO1 played critical functions on the periphery of Cx43-containing GJ plaques forming part of the 'perinexus' to facilitate newly inserted hemichannels at each side of the junction (*Rhett and Gourdie, 2012*). Further, the ZO1/Cx43 interaction was critical for GJ communication before the channel 'ages' and is subsequently removed during channel turnover, a process governed by Cx43 phosphorylation (*Laird, 1996*; *Laird, 2006*; *Márquez-Rosado et al., 2012*; *Solan and Lampe, 2016*; *Thévenin et al., 2017*). However, preventing the ZO1/Cx43 interaction does not prevent GJ formation (*Hunter and Gourdie, 2008*; *Hunter et al., 2003*; *Hunter et al., 2005*). By contrast, our results demonstrated that ZO1b's presence was asymmetric and required for robust Connexin localization and synaptic function. Beyond our observations here, ZO1 is likely to have homeostatic functions in the modulation of Connexin usage at electrical synapses. For example, analysis of the interactions between ZO1 and Cx36 (and its fish homologs) revealed the interactions occurs via a different PDZ domain than the interaction with Cx43 and the interaction with Cx36 has lower affinity and faster kinetics than that of Cx43 (*Flores et al., 2008*; *Li et al., 2004*). This suggests ZO1 has a more dynamic interaction with neuronal Connexins, which may serve the plastic, activity-dependent regulation of electrical transmission observed in fish and mammalian electrical synapses (*Haas et al., 2011*; *Landisman and Connors, 2005*; *Mathy et al., 2014*; *Pereda and Faber, 1996*; *Pereda et al., 1998*; *Turecek et al., 2014*; *Yang et al., 1990*). Thus, our findings suggest that ZO1's function at electrical synapses differs from its role at Cx43-containing GJs, perhaps serving a specialized function in synaptic communication.

## Electrical and chemical synapse coordination

Our results highlight that neural circuit function requires functional electrical and chemical synapses to create an appropriate behavioral response. Our data indicates that the lack of electrical transmission in ZO1b and Connexin mutants did not prevent the formation of co-existing glutamatergic synapses at CEs on the Mauthner cell. The remaining chemical transmission supported a behavioral response organized by the Mauthner-cell network, although importantly, the response showed deficits in performance and altered sensitivity. Given that the startle behavior mediates predator avoidance, these defects would likely be detrimental to survival (*Hecker et al., 2020*). The lack of effect on glutamatergic transmission contrasts a wealth of data supporting a strong, interdependent relationship between the formation of electrical and chemical synapses during development in both invertebrate and vertebrate nervous systems (*Jabeen and Thirumalai, 2018*). For example, in the leech, knockdown of Innexins at a developmental stage where synaptic contacts are solely electrically coupled prevents the formation of later-forming chemical transmission (*Todd et al., 2010*). Similarly, in the developing mouse neocortex, dominant negative constructs of Cx26 prevent the initial electrical coupling and subsequent chemical synapse formation amongst sister excitatory neurons within ontogenetic columns (*Yu et al., 2012*). Our findings indicate that this deficit does not occur at zebrafish CEs when Cx35.5, Cx34.1, and ZO1b are removed. Whether this is due to the differences in the GJ proteins used in the Mauthner cell or instead due to mechanisms that specify CE formation, remains unknown. One intriguing possibility is that other proteins may act as a common synaptic control mechanism that is independent of GJ-forming proteins. Indeed, mutations in the scaffold Neurobeachin caused parallel defects in the formation of both electrical and chemical synapses of the Mauthner circuit (*Miller et al., 2015*), yet the mechanism by which this coordination occurs remains to be elucidated. Alternatively, rather than functional (conductive) GJ channels, other structural components of the electrical synapse may be sufficient to trigger chemical synapse formation via protein-protein interactions. We posit such interactions would apply to mammalian electrical synapses, which can co-exist with neighboring glutamatergic synapses at distances comparable to those found in fish mixed synapses (*Nagy et al., 2018*) and may mediate similar functional interactions.

## The 'electrical synapse density'

Identifying the functional relevance of an intracellular scaffolding protein as a critical part of the electrical synapse draws parallels with our understanding of chemical transmission, where neurotransmitter receptors are clustered and modified by a rich network of postsynaptic proteins that dynamically shape synaptic structure and function. This chemical synapse protein network is known as the 'postsynaptic density' (PSD), a term resulting from its structural identification by EM (*Cohen, 2013*; *Palay, 1956*) and is composed of hundreds of unique proteins (*Grant, 2019*). Mirroring those findings, EM images of electrical synapses in mammals (*Llinas et al., 1974*) and fish (*Brightman and Reese, 1969*) revealed the presence of clearly identifiable electrodense structures, first described as 'semi-dense material' by *Sotelo and Korn, 1978*. These electrodense structures are localized intracellularly and form an undercoating band at neuronal GJs. The identified structures are presumably formed by a proteinaceous 'organelle' that resembles that found at PSDs (*Feng et al., 2019*). As evidence grows for an array of proteins localized to electrical synapses, we propose that ZO1 is member of such an organelle.

Our data provide enticing hints that the molecular framework of the electrical synapse extends beyond the ZO/Connexin interaction. In particular, the fact that ZO1 can localize to sites of synaptic contact independent of the Connexins, and that there remains immunofluorescent staining at Mauthner cell synapses in *tjp1b/ZO1b*[-/-] mutants, both imply the existence of additional proteins that contribute to this synaptic structure. We presume the remaining ZO1 staining in *tjp1b/ZO1b*[-/-] mutants comes from either the paralogous ZO1a protein (*tjp1a*) or from the related ZO2 (*tjp2a*, *tjp2b*) and ZO3 (*tjp3*) proteins. However, our previous analysis of ZO2/ZO3 mutants in zebrafish did not reveal overt defects in Connexin localization (*Marsh et al., 2017*), yet both proteins are localized to mammalian electrical synapses (*Li et al., 2009*). Whether these related scaffolds have functional roles at electrical synapses that were undetected in our initial screen remains to be determined. Beyond the ZO-family, other molecules can directly interact with Cx36 and/or localize at mammalian electrical synapses, such as cell adhesion molecules, cytoskeletal interacting proteins, and molecules that regulate Connexin post-translational modifications (*Lynn et al., 2012*; *Martin et al., 2020*; *O'Brien and*

*Bloomfield, 2018*). Yet, the molecular roles of these proteins at the electrical synapse remain poorly defined. Therefore, as opposed to simple aggregates of intercellular channels, we propose that electrical synapses are complex synaptic structures at which communicating pre- and postsynaptic Connexin hemichannels are governed by a yet to be determined mechanism that builds an asymmetric molecular scaffold. Based on its analogy to the known functions of glutamatergic PSDs, we propose to name this organizational organelle the 'electrical synapse density', or 'ESD' (*Lynn et al., 2012*; *Miller and Pereda, 2017*; *Figure 7C*).

## Electrical synapse structural and functional diversity

Chemical synapses are organized to match their specific functional requirements by combining presynaptic release properties with unique combinations of postsynaptic receptors. Electrical synapses also provide a variety of specific synaptic functions, yet we know little about the source of their diversity. Work in *C. elegans* exposed the large variety of Innexin expression in neurons contributing to neural circuits, with dozens of potentially unique cellular combinations that are altered during development and following environmental stress (*Bhattacharya et al., 2019*). These observations have greatly increased the appreciation of the incidence and potential functional diversity of electrical transmission in invertebrates. Our results indicate that the complexity of electrical synaptic transmission must also include their molecular scaffolds. Scaffold diversity may be more relevant for the function of vertebrate electrical synapses, which in contrast to *C. elegans*, are formed by a smaller number of GJ-forming proteins, with most electrical communication being reliant on Cx36-related proteins investigated here. By promoting and regulating channel trafficking and regulatory molecules, ZO1, and other scaffolding proteins, could support a variety of functions by governing channel function and the local regulatory environment. Thus, future investigations will further expose the molecular composition and functional roles of ZO1 and the ESD. Unraveling the functional complexity of the ESD will lead to a deeper understanding of the diversity of the molecular organization underlying electrical transmission and its contributions to brain function.

# Materials and methods

## Zebrafish

Fish were maintained in the University of Oregon's and the Albert Einstein College of Medicine fish facilities with approval from Institutional Animal Care and Use Committees of each institution. Zebrafish, *Danio rerio*, were bred and maintained at 28°C on a 14 hr on and 10 hr off light cycle. Animals were housed in groups, generally of 25 animals per tank. Development time points were assigned via standard developmental staging (*Kimmel et al., 1995*). All fish used for this project were maintained in the ABC background developed at the University of Oregon. Most fish had the enhancer trap transgene *zf206Et (M/CoLo:GFP)* in the background (*Satou et al., 2009*), unless otherwise noted. Mutant lines were genotyped for all experiments. All immunohistochemistry, electrophysiological, and behavioral experiments were performed at 5 dpf. At this stage of development, zebrafish sex is not yet determined (*Wilson et al., 2014*). Protein extractions were performed from both male and female adult brains and combined.

## Cell culture

HEK293T/17 verified cells were purchased from ATCC (CRL-11268; STR profile, amelogenin: X). Cells were expanded and maintained in Dulbecco's Modified Eagle's Medium (DMEM, ATCC) plus 10% fetal bovine serum (FBS, Gibco) at 37°C in a humidified incubator in the presence of 5% $CO_2$. Low passage aliquots were cryopreserved and stored according to manufacturer's instructions. Cells from each thawed cryovial were monitored for mycoplasma contamination using the Universal Mycoplasma Detection Kit (ATCC, 30–1012K).

## Immunohistochemistry and confocal imaging

Anesthetized, 5–6 days post fertilization (dpf) larvae were fixed for 3 hr in 2% trichloroacetic acid in PBS. Fixed tissue was washed in PBS + 0.5% Triton X-100, followed by standard blocking and antibody incubations. Primary antibody mixes included combinations of the following: rabbit anti-Cx35.5 (Fred Hutch Antibody Technology Facility, clone 12H5, 1:800), mouse IgG1 anti-Cx35.5 (Fred Hutch

Antibody Technology Facility, clone 4B12, 1:250), rabbit anti-Cx34.1 (Fred Hutch Antibody Technology Facility, clone 3A4, 1:250), mouse IgG2A anti-Cx34.1 (Fred Hutch Antibody Technology Facility, clone 5C10A, 1:350), mouse IgG1 anti-ZO1 (Invitrogen, 33–9100, 1:350), mouse IgG2a anti-V5 peptide (Invitrogen, R960-25, 1:50), and chicken anti-GFP (abcam, ab13970, 1:350- 1:500). All secondary antibodies were raised in goat (Invitrogen, conjugated with Alexa-405,–488, −555, or −633 fluorophores, 1:500). Tissue was then cleared stepwise in a 25%, 50%, 75% glycerol series, dissected, and mounted in ProLong Gold antifade reagent (ThermoFisher, P36930). Images were acquired on a Leica SP8 Confocal using a 405-diode laser and a white light laser set to 499, 553/554/557 (consistent within experiments), and 631 nm, depending on the fluorescent dye imaged. Each laser line's data was collected sequentially using custom detection filters based on the dye. Quantitative images of the Club Endings (CEs) were collected using a 63x, 1.40 numerical aperture (NA), oil immersion lens, and images of M/Colo synapses were collected using a 40x, 1.20 NA, water immersion lens. For each set of images, the optimal optical section thickness was used as calculated by the Leica software based on the pinhole, emission wavelengths, and NA of the lens. Within each experiment where fluorescence intensity was to be quantified, all animals (including 3–5 wildtype controls) were stained together with the same antibody mix, processed at the same time, and all confocal settings (laser power, scan speed, gain, offset, objective, and zoom) were identical. Multiple animals per genotype were analyzed to account for biological variation. To account for technical variation, fluorescence intensity values for each region of each animal were an average across multiple synapses.

For high-contrast imaging of the CEs, fixed samples were washed three times with PBS, incubated at 4℃ overnight, and hindbrains were dissected out. Dissected hindbrains were washed, blocked, and stained as above with the following primary antibodies: rabbit anti-Cx35.5 (12H5, 1:500), mouse anti-Cx35/36 (EMD Millipore, MAB3045, 1:250), mouse IgG2A anti-Cx34.1 (5C10A, 1:200), mouse IgG1 anti-ZO1 (33–9100, 1:200), rabbit anti-GFP (Invitrogen, G10362, 1:200), and chicken anti-GFP (Invitrogen, A10262, 1:200). Secondary antibodies were raised in goat (Invitrogen, conjugated with Alexa-405,–488, −546,–555, −633, or −647 fluorophores, 1:200). Samples were then transferred onto a slide in the dark and mounted with Fluoromount-G (Southern Biotech, 0100–01), covered using the standard 'bridge' procedure, and sealed with nail polish. Samples were imaged on LSM 710 and LSM 880 Zeiss microscopes using appropriate laser wavelengths and detection filters. Image stacks of roughly 20 µm were collected using a 63x, 1.40 numerical aperture (NA), oil immersion lens. For each Mauthner, laser strengths and gains were adjusted to achieve maximum visualization of CE staining.

## Electrophysiology

Electrophysiological responses were obtained during whole-cell recordings of Mauthner cells (M-cells) in wt, Cx and ZO-1 mutant zebrafish larvae (5–7 dpf). For this purpose, fish were first anesthetized with a 0.03% solution of MS222 (pH adjusted to 7.4 with NaHCO$_3$) and later transferred to external solution containing d-tubocurarine (10 µM, Sigma). The external solution (in mM): 134 NaCl, 2.9 KCl, 2.1 CaCl$_2$, 1.2 MgCl$_2$, 10 HEPES, 10 Glucose, pH adjusted to 7.8 with NaOH (*Yao et al., 2014*). Zebrafish larvae were put on their backs onto a Sylgard-coated small culture dish (FluoroDish, WPI) and kept in place using fine tungsten pins. The hindbrain was then exposed ventrally following the dissection approach previously described (*Koyama et al., 2011*). Following this procedure, the larvae were placed on an Axio Examiner upright microscope (Carl Zeiss AG) equipped with a recording set-up and superfused with external solution during the entire recording session. The M-cells were identified by GFP expression and/or far-red DIC optics. Patch pipettes (3–4 MΩ) were filled with internal solution (in mM): 105 K-Methanesulfonate, 10 HEPES, 10 EGTA, 2 MgCl$_2$, 2 CaCl$_2$, 4 Na$_2$ATP, 0.4 Tris-GTP, 10 K$_2$-Phosphocreatine, 25 mannitol, pH adjusted to 7.2 with KOH. Whole-cell recordings under the current-clamp configuration were performed with a Multiclamp 700B amplifier and a Digidata 1440A (Molecular Devices) digitizer. The liquid-liquid junction potential was estimated in −16 mV using Clampex 10.6 (Molecular Devices) and was subtracted from the measured values. The electrode's resistance was compensated using the bridge balance feature of the amplifier. To activate the auditory afferents terminating as CEs on the M-cell, a septated (theta) glass pipette was filled with external solution and positioned near the posterior macula of the ear, where the dendritic processes of auditory afferents contact the hair cells (*Yao et al., 2014*). The maximal amplitude of the electrical and chemical components was estimated by applying shocks of increasing intensity until the amplitude of the electrical component did not further increase and before

additional responses with longer latency were evoked. To estimate the Paired-Pulse Ratio (PPR) of the chemical component one, a stimulating-pulse was applied to record 10–20 traces in basal conditions. Then two-stimulating pulses (2 ms apart) were applied to record (10–20 traces) facilitation of the chemical component. Traces that clearly showed a chemical component were averaged for basal and facilitated conditions. The trace in basal conditions was subtracted from the facilitated trace. The PPR was then calculated using the amplitude of the chemical component of the facilitated trace divided by the amplitude of the chemical component in basal conditions. Spontaneous electrical and chemical synaptic responses were assessed during offline analysis of continuous (10 s long) recordings using Clampfit (Axon instruments) to automatically identify events based on their duration (<1.1 ms for electrical spontaneous events and >1.1 ms for chemical spontaneous events). Potential erroneous identification of spontaneous events by the software was monitored manually by verifying the duration of the events. The assignment of electrical vs. chemical nature of spontaneous synaptic events by their duration was confirmed pharmacologically. For synaptic transmission blockade, CNQX was first dissolved in DMSO to have a stock solution of 10 mM. The pharmacological agents used to block synaptic transmission were added to the external solution: Meclofenamic Acid (200 µM, Sigma), CNQX and DAP5 (20 µM, Tocris Biosciences). The M-cell input resistance was estimated by applying a hyperpolarizing-current step of −1 nA and 20 ms in duration and measuring the voltage deflection caused, followed by derivation of resistance with Ohm's law. The rheobase, defined as the minimum depolarizing current of infinite duration necessary to evoke an action potential, was determined by delivering a 20 ms current pulse of increasing intensity. Voltage threshold for action potential generation was determined by applying a depolarizing-current step.

## Behavioral analysis

Startle behavior of 5dpf larvae was analyzed as described previously (*Marsden et al., 2018*). Briefly, larvae were adapted to the testing temperature and lighting conditions for 30 min and then transferred to individual wells of a custom, laser-cut acrylic multi-well testing arena, illuminated from below with an infrared (IR) LED array and from above with a white light LED bulb to simulate daylight conditions. A total of 60 acoustic stimuli, 10 at each of 6 intensities, were delivered pseudorandomly using an acoustic-vibrational shaker (Bruel and Kjaer) with an inter-stimulus interval of 20 s to eliminate habituation to repeated stimulation (*Wolman et al., 2015*). The intensity of each stimulus was calibrated using a PCB Piezotronics accelerometer (model #355B04) and signal conditioner (model #482A21), and voltage outputs were converted to dB using the formula dB = 20 log (V/0.775). Behavioral responses were captured at 1000 frames per second with an IR-sensitive Photron mini-UX50 high-speed camera. After testing, larvae were fixed in methanol for subsequent genotyping, thus all testing and analysis was performed blind to genotype. Behavioral responses were tracked and analyzed using FLOTE software (*Burgess and Granato, 2007*), with short and long latency C-bend responses (SLCs and LLCs, respectively) automatically defined based on the kinematic parameters of the response. Startle sensitivity index was calculated by measuring the area under the curve of stimulus intensity versus SLC frequency for each larva.

## Cell transfection and immunoprecipitation

Cell lines were obtained from ATCC, identity ensured by using exclusively low-passage cells, and were confirmed to be mycoplama free. Full-length Cx34.1 and full-length Cx35.5 were cloned into the pCMV expression vector. Full-length ZO1b was cloned into the pCMV expression vector with an $NH_2$-terminal mVenus tag and a COOH-terminal 8xHIS tag. Low passage HEK293T/17 cells were seeded 24 hr prior to transfection ($1 \times 10^6$ cells/well of a six-well dish), and the indicated plasmids were co-transfected using Lipofectamine 3000 (Invitrogen) following the manufacturer's instructions. Cells were collected 36–48 hr post-transfection and lysed in 0.25 ml solubilization buffer (50 mM Tris [pH7.4], 100 mM NaCl, 5 mM EDTA, 1.5 mM MgCl2, 1 mM DTT and 1% Triton X-100) plus a protease inhibitor cocktail (Pierce). Lysates were centrifuged at 20,000 x g for 30 min at 4°C, and equal amounts of extract were immunoprecipitated with 0.5 ug rabbit anti-GFP (Abcam, Ab290) overnight with rocking at 4°C. Immunocomplexes were captured with 25 µl prewashed Protein A/G agarose beads for 1 hr with rocking at 4°C. Beads were washed three times with lysis buffer, and bound proteins were boiled for 3 min in the presence of LDS-PAGE loading dye containing 200 mM DTT. Samples were resolved by SDS-PAGE using a 4–15% gradient gel and analyzed by Western blot using

the following primary antibodies: rabbit anti-GFP (Abcam Ab290), rabbit anti-Cx34.1 3A4-conjugated-680LT, and mouse anti-Cx35.5 4B12. Compatible near-infrared secondary antibodies were used for visualization with the Odyssey system (LI-COR).

## Immunoprecipitation of fish brain homogenates

Brains from *wildtype*, *tjp1a/ZO1a⁻/⁻*, or *tjp1b/ZO1b⁻/⁻* euthanized adult fish (4–15 months old) were removed, snap frozen in liquid nitrogen and stored at −80C until use. Brains were homogenized in 1 ml of HSE buffer (20 mM Hepes [pH7.5], 150 mM NaCl, 5 mM EDTA, 5 mM EGTA, and 1 mM DTT) plus a protease inhibitor cocktail using a glass homogenizer. Detergent was added to the homogenate (final 2% octyl ß-D-glucopyranoside, Anatrace) and solubilized overnight with rocking at 4°C. Solubilized homogenate was cleared by centrifugation at 20,000 x g for 30 min at 4°C, then pre-cleared for 1 hr at 4°C with Protein A/G beads before immunoprecipitation. The protein concentration for each homogenate was measured by Bradford assay. Pre-cleared homogenates (2 mg/IP) were immunoprecipitated with 0.5 µg mouse anti-ZO1 (Life Technologies, 33–9100), control mouse IgG (Jackson ImmunoResearch), mouse anti-Cx34.1 5C10, or mouse anti-Cx35.5 4B12 antibody overnight with rocking at 4°C. Immunocomplexes were captured with 25 ul prewashed Protein A/G agarose beads for 1 hr with rocking at 4°C. Beads were washed three times with HSE buffer, and bound proteins were boiled for 3 min in the presence of LDS-PAGE loading dye containing 200 mM DTT. Immune complexes were examined by western analysis using the following primary antibodies: mouse anti-ZO1, rabbit anti-Cx34.1 3A4-conjugated-680LT, rabbit anti-Cx35.5 12H5-conjugated-680LT, or mouse anti-Cx35.5 4B12. Compatible near-infrared secondary antibodies were used for visualization.

## Bacterial expression and purification of proteins

The Cx34.1-tail (aa256-299), Cx34.1-tail ΔPBM (aa256-295), Cx35.5-tail (aa267-309), and Cx35.5-tail ΔPBM (aa267-305) were cloned into the pGEX expression vector allowing for an $NH_2$-terminal GST tag. ZO1b-PDZ1 (aa105-207) and ZO1b-PDZ2 (aa298-387) were cloned into a modified pET expression vector (pBH) to allow for an $NH_2$-terminal 6xHis tag followed by a TEV cleavage site (vectors kindly provided by Ken Prehoda). Plasmids were transformed in *E. coli* BL21(DE3) cells and plated on selective LB plates. Single colonies were picked to inoculate 2 ml starter cultures and grown overnight. Overnight cultures were inoculated into 250 ml selective LB and grown for ~3 hr at 37°C with shaking until OD600 reached 0.8–1 followed by 4 hr induction with 0.4 mM IPTG. Cell pellets were collected by centrifugation at 6000 RPM for 5 min at 4°C and frozen at −20°C until test samples confirmed expression. Pellets were resuspended in sonication buffer (50 mM NaPO4 [pH7.4], 300 mM NaCl, and 1 mM PMSF). After adding a dash of lysozyme, the mixture was incubated on ice for 30 min. Resuspended bacteria were sonicated on ice at 50% amplitude, 1 s/1 s pulse on/off, four times for 20 s. Debris was cleared by centrifugation at 16,000 x g for 30 min at 4°C. For GST fusions, supernatant was added to 200 ul pre-washed glutathione agarose resin and incubated overnight with rocking at 4°C. Beads were washed three times with sonication buffer and stored at 4°C. Purity and amount loaded onto resin was determined by SDS-PAGE followed by Coomassie stain. For 6xHIS fusions, supernatant was brought to a final concentration of 20 mM imidazole and incubated with pre-washed His60 resin overnight with rocking at 4°C. Resin was washed with sonication buffer containing 20 mM imidazole. Protein was eluted from the resin with sonication buffer containing 250 mM imidazole. The protein was concentrated and exchanged into imidazole-free buffer using an Amicon centrifugal filter unit (10K MWCO) and stored at 4°C on ice. Protein concentration was estimated by A205 (https://spin.niddk.nih.gov/clore/) (*Anthis and Clore, 2013*), and purity was determined by SDS-PAGE followed by Coomassie stain.

## In vitro binding assay

Equal amounts of GST fusions (10 µl bed of resin) were aliquoted and the storage buffer was removed. To each sample 15 µl of 6xHIS-ZO1b-PDZ1 (7 mg/ml) was added, gently mixed and incubated at room temperature for 15 min. Resin was washed three times with cold wash buffer (50 mM $NaPO_4$ [pH7.4], 300 mM NaCl). After the last wash, all buffer was removed and resin was resuspended in 10 µl LDS-PAGE dye with 200 mM DTT. Samples were boiled for 3 min and resolved by SDS-PAGE using a 4–20% gradient gel. Samples were analyzed by Western blot using rabbit anti-

TEV cleavage site primary antibody (ThermoFisher, PA1-119) and visualized with a compatible near-infrared secondary antibody. A portion of the GST fusion resin was analyzed by Coomassie stain to demonstrate equal loading.

## In vitro overlay assay

Equal amounts of GST fusion were resuspended in LDS-PAGE dye plus 200 mM DTT, boiled for 3 min, resolved by SDS-PAGE on a 4–15% gradient gel, transferred to nitrocellulose, and blocked with 5% milk in TBS (20 mM Tris [pH7.4], 150 mM NaCl). Blocked membranes were incubated with TBS-T buffer (TBS + 0.1% Tween-20) alone (mock overlay), or TBS-T containing 1 µM 6xHIS-ZO1b-PDZ1 or 1 uM 6xHIS-ZO1b-PDZ2 overnight with shaking at 4°C. Membranes were processed for western analysis using rabbit anti-TEV cleavage site primary antibody and a compatible near-infrared secondary antibody to detect bound PDZ protein. Equal loading of GST fusions was determined by Stain-Free imaging technology.

## Cas9-mediated genome engineering of V5-tjp1b transgenics

A single guide RNA (sgRNA) targeting the 5' region of the endogenous *tjp1b* coding sequence (sequence in Key Resources Table) was designed using the CRISPRscan algorithm (*Moreno-Mateos et al., 2015*) and synthesized as previously described (*Shah et al., 2015*). The sgRNA was generated using the T7 megascript kit (ThermoFisher, AMB13345). The V5-*tjp1b* single stranded donor oligo (ssODN) was designed to repair into the endogenous *tjp1b* locus and was synthesized complimentary to the coding strand. The ssODN contained 40 bp homology arms which flanked an XbaI restriction site, V5 sequence, and a 5x glycine linker, respectively (sequence in Key Resources Table). Upon correct repair, the inserted sequence was designed to disrupt the endogenous sgRNA recognition site to prevent further double stranded breaks after repair. Injection mixes were prepared in a pH 7.5 buffer solution of 300 mM KCl and 4 mM HEPES and contained a final concentration of 200 pg/nL ssODN, 200 pg/nL gRNA, and 1600 pg/nL Cas9 protein (IDT, 1081058). Injection mixes were incubated at 37 °C for 5 min immediately prior to injection to promote formation of the Cas9 and sgRNA complex. Finally, 1 nL of solution was injected into embryos at the one-cell stage. Injected embryos were raised to adulthood and outcrossed to wild-type animals. Animals carrying insertions were identified and verified using PCR and Sanger sequencing.

## Blastula cell transplantation

Cell transplantation was performed at the high stage approximately 3.3 hr into zebrafish development using standard techniques (*Kemp et al., 2009*). Embryos were chemically de-chorionated with protease (Sigma Aldrich, 9036-06-0) prior to transplantation. Cells were transplanted using a 50 mm wide glass capillary needle attached to an oil hydraulic. For 'V5-*tjp1b+/-* into *wildtype*' transplants (*Figures 7C, D* and *Figure 7—figure supplement 1B-H*) cells from animals heterozygous for *V5-tjp1b* in the M/CoLo:GFP background were transplanted into non-transgenic *wildtype* hosts. For '*tjp1b-/-* into *wildtype*' transplants (*Figure 7G and H*), genotyped animals homozygous for the *tjp1b*$^{\Delta16bp}$ mutation in the M/CoLo:GFP background were crossed and progeny were transplanted into non-transgenic *wildtype* hosts. For '*wildtype* into *tjp1b-/-*' transplants (*Figure 7I and J*), transgenic *M/CoLo:GFP wildtype* animals were crossed to use as donors, and non-transgenic, homozygous *tjp1b*$^{\Delta16bp}$ animals were crossed to produce hosts. Approximately 20 cells were deposited ~10–15 cell diameters away from the margin, with a single donor embryo supplying cells to 3–5 hosts. At 5 dpf, larvae were fixed in TCA and processed for immunohistochemistry.

## Analysis of confocal imaging

For fluorescence intensity quantitation, confocal images were processed and analyzed (in part or in full) using FiJi (*Schindelin et al., 2012*) software. To quantify staining at M/Colo synapses, a standard region of interest (ROI) surrounding each M/CoLo site of contact was drawn, and the mean fluorescence intensity was measured. For the quantification of staining at the club endings, confocal z-stacks of the Mauthner soma and lateral dendrite were cropped to 36.08 µm x 36.08 µm centered around the lateral dendritic bifurcation. Using the SciPy (*Virtanen et al., 2020*) and scikit-image (*van der Walt et al., 2014*) computing packages, the cropped stack was then cleared outside of the Mauthner cell, a 3$^3$ median filter was applied to reduce noise, and a standard threshold was set

within each experiment to remove background staining. The image was then transformed into a max intensity projection and the integrated density of each stain within the Mauthner cell was extracted. Where counts were used to quantify antibody labeling of CEs in high-contrast images, CEs were identified as large fluorescently labeled oval areas of approximately 1.5–2 microns on the distal portion and fork of the Mauthner cell's lateral dendrite, which were thoroughly examined with the confocal microscope. For *Figure 7—figure supplement 1I, J*, the presence or absence of electrical synapses on Mauthner was quantified as counts of stereotyped electrical synapse structures dually labeled for Cx34.1 and Cx35.5.

Standard deviations and errors were computed using Prism (GraphPad) or Excel (Microsoft) software. Figure images were created using FiJi, Photoshop (Adobe), and Illustrator (Adobe). Statistical analyses were performed using Prism (GraphPad) and either an unpaired t-test with Welch's correction or a one-way analysis of variance with Bonferrroni's multiple comparison test was performed. For all experiments, values were normalized to *wildtype* control animals, and n represented the number of fish used. Fish were excluded from analysis if Mauthner morphology/GFP staining was abnormal.

### Analysis of electrophysiological recordings

Electrophysiology traces of spontaneous synaptic events were analyzed using Clampfit 10.7 software. Electrophysiology traces were transferred to Canvas for illustration and to OriginPro software for quantification and statistical analyses. For the different parameters measured using electrophysiological recordings, means and standard error of the mean (SEM) were illustrated using Prism (GraphPad) and computed using OriginPro software. For statistical analyses comparing electrophysiological recordings in wildtype and mutant zebrafish, Mann-Whitney and Kruskal-Wallis nonparametric tests were performed using OriginPro software.

## Acknowledgements

We thank members of the Pereda and Miller lab for ongoing support, comments, and discussions on this manuscript. We thank members of the Albert Einstein College of Medicine and the University of Oregon for critical feedback on this work. We thank the University of Oregon AqACS facility for superb animal care. This work was supported by and a NIH Eunice Kennedy Shriver National Institute of Child Health and Human Development (NICHD), Developmental Biology Training Grant T32HD007348 to AML, an NICHD Ruth L Kirschstein National Research Service Award F32HD102182 to EAM, NIH grants R01DC011099 from the National Institute on Deafness and Other Communication Disorders (NIDCD) and R21NS085772 from the National Institute of Neurological Disorders and Stroke (NINDS) to AEP, RF1MH120016 from the National Institutes of Mental Health (NIMH) to ACM and AEP, and an R21NS117967 and R01NS105758 from the NINDS to ACM.

## Additional information

### Funding

| Funder | Grant reference number | Author |
| --- | --- | --- |
| Eunice Kennedy Shriver National Institute of Child Health and Human Development | T32HD007348 | Abagael M Lasseigne |
| Eunice Kennedy Shriver National Institute of Child Health and Human Development | F32HD102182 | E Anne Martin |
| National Institute on Deafness and Other Communication Disorders | R01DC011099 | Alberto Pereda |
| National Institute of Neurological Disorders and Stroke | R21NS085772 | Alberto Pereda |
| National Institute of Mental Health | RF1MH120016 | Alberto Pereda Adam C Miller |

| National Institute of Neurological Disorders and Stroke | R21NS117967 | Adam C Miller |
| National Institute of Neurological Disorders and Stroke | R01NS105758 | Adam C Miller |
| Eunice Kennedy Shriver National Institute of Child Health and Human Development | | Abagael M Lasseigne |

The funders had no role in study design, data collection and interpretation, or the decision to submit the work for publication.

## Author contributions

Abagael M Lasseigne, Conceptualization, Data curation, Formal analysis, Funding acquisition, Validation, Investigation, Visualization, Methodology, Writing - original draft, Writing - review and editing; Fabio A Echeverry, Data curation, Formal analysis, Validation, Investigation, Visualization, Methodology, Writing - original draft, Writing - review and editing; Sundas Ijaz, Data curation, Formal analysis, Validation, Investigation, Visualization, Methodology, Writing - review and editing; Jennifer Carlisle Michel, Conceptualization, Data curation, Formal analysis, Validation, Investigation, Visualization, Methodology, Writing - original draft, Writing - review and editing; E Anne Martin, Data curation, Formal analysis, Funding acquisition, Validation, Investigation, Visualization, Methodology, Writing - review and editing; Audrey J Marsh, Data curation, Formal analysis, Validation, Investigation, Methodology, Writing - review and editing; Elisa Trujillo, Data curation, Formal analysis, Validation, Investigation, Visualization, Methodology; Kurt C Marsden, Investigation, Methodology, Writing - review and editing; Alberto E Pereda, Adam C Miller, Conceptualization, Resources, Data curation, Formal analysis, Supervision, Funding acquisition, Validation, Investigation, Visualization, Methodology, Writing - original draft, Project administration, Writing - review and editing

## Author ORCIDs

Fabio A Echeverry (iD) https://orcid.org/0000-0002-4200-4080
Kurt C Marsden (iD) http://orcid.org/0000-0001-8087-6181
Alberto E Pereda (iD) https://orcid.org/0000-0002-8283-8768
Adam C Miller (iD) https://orcid.org/0000-0001-7519-3677

## Ethics

Animal experimentation: This study was performed in strict accordance with the recommendations in the Guide for the Care and Use of Laboratory Animals of the National Institutes of Health. All of the animals were handled according to approved institutional animal care and use committee (IACUC) protocols (#AUP-18-35) of the University of Oregon.

## Decision letter and Author response

Decision letter https://doi.org/10.7554/eLife.66898.sa1
Author response https://doi.org/10.7554/eLife.66898.sa2

# Additional files

## Supplementary files

- Transparent reporting form

## Data availability

All data generated or analyzed during this study are included in the manuscript and supporting files. Source data files have been provided for all figures. Zebrafish lines are available at the Zebrafish International Resource Center and/or via contacting the lead author.

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

# Appendix 1

**Appendix 1—key resources table**

| Reagent type (species) or resource | Designation | Source or reference | Identifiers | Additional information |
|---|---|---|---|---|
| Gene (*Danio rerio*) | gjd1a | ZFIN | ZDB-GENE-080723–77 | |
| Gene (*Danio rerio*) | gjd1b | ZFIN | ZDB-GENE-100921–89 | |
| Gene (*Danio rerio*) | gjd2a | ZFIN | ZDB-GENE-111020–17 | |
| Gene (*Danio rerio*) | gjd2b | ZFIN | ZDB-GENE-030911–1 | |
| Gene (*Danio rerio*) | tjp1a | ZFIN | ZDB-GENE-031001–2 | |
| Gene (*Danio rerio*) | tjp1b | ZFIN | ZDB-GENE-070925–1 | |
| Strain, strain background (*Danio rerio*) | AB x Tübingen | University of Oregon fish facility | ZFIN: ZDB-GENO-010924–10; PubMed: PMC4667794 | |
| Strain, strain background (*Danio rerio*) | M/CoLo:GFP (zf206Et) | *Satou et al., 2009* | ZFIN: ZDB-ALT-110217–6; PubMed: 19474306 | |
| Strain, strain background (*Danio rerio*) | tjp1b$^{\Delta16bp}$ (fh448) | *Shah et al., 2015* | ZFIN: ZDB-ALT-160825–6; PubMed: PMC4667794 | |
| Strain, strain background (*Danio rerio*) | tjp1a$^{\Delta2bp}$ (fh463) | *Marsh et al., 2017* | ZFIN: ZDB-ALT-180920–6; Pubmed: PMC5698123 | |
| Strain, strain background (*Danio rerio*) | gjd1a$^{dis3}$ (fh360) | *Miller et al., 2017* | ZFIN: ZDB-ALT-160825–2: Pubmed: PMC5462537 | |
| Strain, strain background (*Danio rerio*) | gjd2a$^{\Delta5bp}$ (fh437) | *Shah et al., 2015* | ZFIN: ZDB-TALEN-170822–4; Pubmed: PMC4667794 | |
| Strain, strain background (*Danio rerio*) | gjd1a$^{\Delta8bp}$ (fh436) | *Shah et al., 2015* | ZFIN: ZDB-TALEN-170822–3; Pubmed: PMC4667794 | |
| Strain, strain background (*Danio rerio*) | V5-tjp1b (b1406) | This paper | N/A | See Materials and methods |
| Strain, strain background (*Homo sapiens*) | HEK293T/17 cells | ATCC | CRL-11268 | |
| Strain, strain background (*Escherichia coli*) | BL21(DE3) | New England BioLabs | C2527I | |
| Strain, strain background (*Escherichia coli*) | DH5alpha | Zymo Research | T3009 | |
| Antibody | Chicken monoclonal anti-GFP IgY | Abcam | ab13970; RRID:AB_300798 | (1:500) |

*Continued on next page*

*Appendix 1—key resources table continued*

| Reagent type (species) or resource | Designation | Source or reference | Identifiers | Additional information |
|---|---|---|---|---|
| Antibody | Rabbit monoclonal anti-GFP | Abcam | Ab290 | (1:1000) |
| Antibody | Mouse monoclonal IgG1 anti-ZO1 | ThermoFisher | 33–9100; RRID: AB_2533147 | (1:350) |
| Antibody | Rabbit monoclonal anti-Cx35.5 | *Miller et al., 2015* | clone 12H5 | (1:800) |
| Antibody | Rabbit monoclonal anti-Cx35.5-IRDye 680LT conjugated | Fred Hutch Antibody Technology Facility, Miller lab conjugated | clone 12H5 | (1:1000) |
| Antibody | Mouse monoclonal IgG1 anti-Cx35.5 | *Miller et al., 2015* | clone 4B12a | (1:1000) |
| Antibody | Rabbit monoclonal anti-Cx34.1 | *Miller et al., 2015* | clone 3A4 | (supernatant) |
| Antibody | Rabbit monoclonal anti-Cx34.1-680LT conjugated | Fred Hutch Antibody Technology Facility, Miller lab conjugated | clone 3A4 | (supernatant) |
| Antibody | Mouse monoclonal IgG2A anti-Cx34.1 | *Miller et al., 2015* | clone 5C10A | (1:200) |
| Antibody | Rabbit monoclonal anti -GluR2/3 | Millipore Sigma | 07–598; RRID:AB_11213931 | (1:250) |
| Antibody | Rabbit monoclonal anti-TEV Cleavage Site | Invitrogen | PA1-119; RRID: AB_2539888 | (1:2000) |
| Antibody | Goat monoclonal anti-rabbit Alexa 405 | Invitrogen | A31556; RRID:AB_221605 | (1:500) |
| Antibody | Donkey monoclonal anti-chicken IgGY Alexa 488 | Jackson Immuno Research Laboratories | 703-545-155 | (1:500) |
| Antibody | Goat monoclonal anti-mouse IgG2a Alexa 555 | Invitrogen | A21137 | (1:500) |
| Antibody | Goat monoclonal anti-mouse IgG1 Alexa 633 | Invitrogen | A21126 | (1:500) |
| Antibody | Mouse monoclonal IgG2a anti-V5 peptide | Invitrogen | R960-25 | (1:1000) |
| Antibody | ChromPure mouse monoclonal IgG, whole molecule | Jackson Immuno Research Laboratories | 015-000-003; RRID: AB_2337188 | (1:1000) |
| Antibody | IRDye 680LT goat monoclonal anti-rabbit secondary | LI-COR | 925–68021; RRID: AB_2713919 | (1:10000) |
| Antibody | IRDye 800CW goat monoclonal anti-mouse secondary | LI-COR | 925–32210; RRID: AB_2687825 | (1:10000) |
| Antibody | Mouse monoclonal IgG kappa binding protein (m-IgGκ BP) conjugated to CruzFluor 790 (CFL 790) secondary | Santa Cruz Biotechnology | sc-516181 | (1:10000) |
| Recombinant DNA reagent | pCMV mammalian expression plasmid | J. O'Brien lab | N/A | |
| Recombinant DNA reagent | pGEX bacterial expression plasmid | K. Prehoda lab | N/A | |

*Continued on next page*

*Appendix 1—key resources table continued*

| Reagent type (species) or resource | Designation | Source or reference | Identifiers | Additional information |
|---|---|---|---|---|
| Recombinant DNA reagent | pBH bacterial expression plasmid | K. Prehoda lab | N/A | |
| Sequence-based reagent | *tjp1b*$^{\Delta16bp}$ genotyping primers: Fwd, TCTCTTTCCTTCT TTCTGTGTGTTT; Rev, AAAAGTGAAATT CTCACCCTGTG | *Marsh et al., 2017* | N/A | |
| Sequence-based reagent | *gjd2a*$^{\Delta5bp}$ genotyping primers: Fwd, GATGAGCAGCG ATGGGAGAAT; Rev, CTTGAATTTCGG CGTCAGACAG | *Miller et al., 2015* | N/A | |
| Sequence-based reagent | *gjd1a*$^{\Delta8bp}$ genotyping primers: Fwd, CTCAGGCTGAAG GTCGGCAGGGAAG; Rev, GCTGTACCGCA GCCTCCAGCAAC | *Miller et al., 2015* | N/A | |
| Sequence-based reagent | *gjd1a*$^{dis3}$ genotyping primers: Fwd, AGTGCGACCGC TACCCTTGC; Rev, AGCACCACGCA GATTCCGCT, | *Miller et al., 2015* | N/A | |
| Sequence-based reagent | *tjp1a*$^{\Delta2bp}$ genotyping primers: Fwd, GTACAACAAT GGAGGAAACTGTCA; Rev, AAAGAAGCTAT GTTCAACACTCACC | *Marsh et al., 2017* | N/A | |
| Sequence-based reagent | *tjp1b* N-terminus Crispr target: GGATTTCT GGTAATTCACCA | This paper | N/A | See Materials and Methods |
| Sequence-based reagent | *tjp1b* N-terminus oligo: GAGCCAGCTGCATAACAGT AATGTATTTCTGGTAA TTCACTCCGCCTCCACC TCCGGTGCTATCCAGGCCCA GCAGCGGGTTCGG AATCGGTTTG CCTCTAGACATG GTACTGTTCAC CGCTTTTTTGAAAC ACAAAAATCCGCA | This paper | N/A | See Materials and Methods |
| Sequence-based reagent | *V5-tjp1b* screening primers: Fwd, GGGAGTAGG AGGAGAAGGA; Rev, GTTTTCTGG GAGGCAGGCTA | This paper | N/A | See Materials and Methods |
| Peptide, recombinant protein | GST-Cx34.1 (aa 256–299) | This paper | GST-Cx34.1-tail wt | See Materials and Methods |
| Peptide, recombinant protein | GST-Cx34.1 (aa 256–295) | This paper | GST- Cx34.1-tail ΔPBM | See Materials and Methods |

*Continued on next page*

*Appendix 1—key resources table continued*

| Reagent type (species) or resource | Designation | Source or reference | Identifiers | Additional information |
|---|---|---|---|---|
| Peptide, recombinant protein | GST-Cx35.5 (aa 267–309) | This paper | GST-Cx35.5-tail wt | See Materials and Methods |
| Peptide, recombinant protein | GST-Cx35.5 (aa 267–305) | This paper | GST- Cx35.5-tail ΔPBM | See Materials and Methods |
| Peptide, recombinant protein | 6xHIS-TEV cleavage site-ZO1b (aa105-207) | This paper | ZO1b PDZ1 | See Materials and Methods |
| Peptide, recombinant protein | 6xHIS-TEV cleavage site-ZO1b (aa 298–387) | This paper | ZO1b PDZ2 | See Materials and Methods |
| Peptide, recombinant protein | mVenus-ZO1b (aa 2–1778)—8xHIS | This paper | mVenus-ZO1b | See Materials and Methods |
| Peptide, recombinant protein | Cx34.1 (aa 1–299) | This paper | Cx34.1-FL | See Materials and Methods |
| Peptide, recombinant protein | Cx34.1 (aa 1–295) | This paper | Cx34.1-ΔPBM | See Materials and Methods |
| Peptide, recombinant protein | Cx35.5 (aa 1–309) | This paper | Cx35.5-FL | See Materials and Methods |
| Peptide, recombinant protein | Cx35.5 (aa 1–305) | This paper | Cx35.5-ΔPBM | See Materials and Methods |
| Commercial assay or kit | Taq 2X Master Mix | NEB | M0270L | |
| Commercial assay or kit | Universal Mycoplasma Detection Kit | ATCC | 30–1012K | |
| Chemical compound, drug | ProLong Gold antifade reagent | ThermoFisher | P36930 | |
| Chemical compound, drug | n-Octyl-β-D-Glucopyranoside, Anagrade | Anatrace | O311 | |
| Chemical compound, drug | Protease Inhibitor Mini Tablets, EDTA-free | Pierce | A32955 | |
| Chemical compound, drug | Lipofectamine 3000 | Invitrogen | L3000008 | |
| Chemical compound, drug | Dulbecco's Modified Eagle's Medium (DMEM) | ATCC | 30–2002 | |

*Continued on next page*

*Appendix 1—key resources table continued*

| Reagent type (species) or resource | Designation | Source or reference | Identifiers | Additional information |
|---|---|---|---|---|
| Chemical compound, drug | Opti-MEM | Gibco | 31-985-062 | |
| Chemical compound, drug | Glutathione resin | Pierce | PI16100 | |
| Chemical compound, drug | His60 Ni Superflow resin | TaKaRa | 635659 | |
| Chemical compound, drug | Protein A/G Agarose | Pierce | PI20421 | |
| Chemical compound, drug | 4–15% Criterion TGX Stain-Free Protein Gel | BioRad | 5678083 | |
| Chemical compound, drug | 4–20% Mini-PROTEAN TGX Stain-Free Protein Gels | BioRad | 4568095 | |
| Chemical compound, drug | (+)-Tubocurarine chloride pentahydrate | Sigma | 93750 | |
| Chemical compound, drug | Ethyl 3-aminobenzoate methanesulfonate salt | Sigma | A5040 | |
| Chemical compound, drug | Sodium chloride | Sigma | 567440 | |
| Chemical compound, drug | Potassium chloride | Sigma | P3911 | |
| Chemical compound, drug | Calcium chloride | Sigma | 21115 | |
| Chemical compound, drug | Magnesium chloride | Sigma | M1028 | |
| Chemical compound, drug | HEPES | Sigma | H3375 | |
| Chemical compound, drug | D-(+)-Glucose | Sigma | G8270 | |
| Chemical compound, drug | Sodium hydroxide | Sigma | S5881 | |
| Chemical compound, drug | Potassium methanesulfonate | Sigma | 83000 | |
| Chemical compound, drug | EGTA | Sigma | E3889 | |
| Chemical compound, drug | Adenosine 5′-phosphosulfate sodium salt | Sigma | A5508 | |

*Appendix 1—key resources table continued*

| Reagent type (species) or resource | Designation | Source or reference | Identifiers | Additional information |
|---|---|---|---|---|
| Chemical compound, drug | Guanosine 5′-triphosphate tris salt | Sigma | G9002 | |
| Chemical compound, drug | Creatine Phosphate, Dipotassium Salt | Sigma | 237911 | |
| Chemical compound, drug | D-Mannitol | Sigma | M4125 | |
| Chemical compound, drug | Potassium Hydroxide | Sigma | P5958 | |
| Chemical compound, drug | Meclofenamic acid sodium salt | Sigma | M4531 | |
| Chemical compound, drug | Dimethyl sulfoxide | Sigma | D8418 | |
| Chemical compound, drug | CNQX | Tocris | 0190 | |
| Chemical compound, drug | DAP-5 | Tocris | 0106 | |
| Chemical compound, drug | Capillary Glass 1.5 mm OD, 1.12 mm ID | WPI | TW150F-3 | |
| Chemical compound, drug | SYLGARD 184 Silicone Elastomer Kit | DOW | https://www.dow.com/en-us/pdp.sylgard-184-silicone-elastomer-kit.01064291z.html | |
| Software, algorithm | GraphPad Prism | Graph Pad Software | https://www.graphpad.com/ | |
| Software, algorithm | Adobe Photoshop CC 2015 | Adobe | https://www.adobe.com/ | |
| Software, algorithm | Adobe Illustrator CC 2015 | Adobe | https://www.adobe.com/ | |
| Software, algorithm | scikit-image | *van der Walt et al., 2014* | https://peerj.com/articles/453/ | |
| Software, algorithm | SciPy | *Virtanen et al., 2020* | https://www.nature.com/articles/s41592-019-0686-2?luicode=10000011&lfid=1008082086c7dfebc09fc300733002ea997ba2_-_feed&u=https%3A%2F%2Fwww.nature.com%2Farticles%2Fs41592-019-0686-2 | |
| Software, algorithm | FiJi | *Schindelin et al., 2012* | PubMed: 22743772; https://fiji.sc/ | |

*Continued on next page*

*Appendix 1—key resources table continued*

| Reagent type (species) or resource | Designation | Source or reference | Identifiers | Additional information |
|---|---|---|---|---|
| Software, algorithm | OriginPro | OriginLab Corp. | https://www.originlab.com/ | |
| Software, algorithm | Clampex | Molecular Devices | | |
| Other | Leica TCS SP8 Confocal | Leica | http://www.leica-microsystems.com/products/confocal-microscopes/details/product/leica-tcs-sp8/ | |
| Other | 40X/1.10 Water Objective | Leica | 11506357 | |
| Other | 63X/1.40 Oil Objective | Leica | 15506350 | |
| Other | Amicon Ultra-0.5 Centrifugal Filter Units, 10K MWCO | MilliporeSigma | UFC501008 | |
| Other | Amicon Ultra-4 Centrifugal Filter Units 10 kDa MWCO | MilliporeSigma | UFC801008 | |
| Other | Upright Axio Examiner Microscope | Carl Zeiss Microscopy, LLC | https://www.zeiss.com/corporate/int/home.html?vaURL=www.zeiss.de/en | |
| Other | 20X/0.5 W-N-Achroplan | Carl Zeiss Microscopy, LLC | 420957–9900 | |
| Other | 40X/1.0 VIS-IR W-Plan-Apochromatic | Carl Zeiss Microscopy | 421462–9900 | |
| Other | Multiclamp 700B amplifier | Molecular Devices | https://www.moleculardevices.com/ | |
| Other | Digidata 1440A | Molecular Devices | https://www.moleculardevices.com/ | |

