## [Decision Letter]

**Acceptance summary:**

We appreciate how you have addressed the reviewer's comments and believe that your paper will make vary valuable contribution to the field of electric synapse biology.

**Decision letter after peer review:**

Thank you for submitting your article "Electrical synaptic transmission requires a postsynaptic scaffolding protein" for consideration by *eLife*. Your article has been reviewed by 2 peer reviewers, and the evaluation has been overseen by a Reviewing Editor and Lu Chen as the Senior Editor. The reviewers have opted to remain anonymous.

The reviewers have discussed their reviews with one another. Due to some disagreements, we need to consult with a third reviewer, which caused some delay for which we apologize. The Reviewing Editor has drafted this to help you prepare a revised submission.

This paper reports a carefully done and well-presented study of the functional relationship between neuronal Connexins and Zonula Occludens 1 (ZO1), an intracellular scaffolding protein localized to electrical synapses. Using model electrical synapses in zebrafish Mauthner cells, the authors demonstrated that ZO1 is required for robust synaptic Connexin localization. Furthermore, by performing elegant chimera experiments, they found that ZO1 is asymmetrically localized exclusively postsynaptically at neuronal contacts where it functions to assemble intercellular channels. The author's findings challenge current perceptions of the functional and molecular organization of electrical synapses and calls for a new model that includes a primary role for the intracellular molecular scaffold in governing the function of intercellular channels.

Only the following revisions need to be implemented:

1. The conclusion that ZO1 plays a role in gap junction regulation/function is not well supported by the data presented here as the authors cannot rule out that fact that gap junctions simply fail to form in ZO1 mutants. The authors should revise their conclusions in results and discussion to tone down the claim.

2. In Figure 2, the authors quantitated ZO1 localization to CEs or M/CoLo and showed that the amount of the localized ZO1 was halved in the connexin mutants. This means that the ZO1 localization to CEs or M/CoLo at least in part depends on connexins. This result should be carefully discussed in the Discussion section. The finding was described in the summarizing sentence on the page 8, "1) ZO1b localizes to putative electrical synaptic sites *largely* independent of Connexin proteins", but was not carefully described in the other sentences. The following sentences should be modified by weakening the asserting phrases in the main texts.

P8

"ZO1b localizes to the electrical synapse *independent of* Cxs"

"ZO1 *does not require* the presence of channel-forming proteins to localize at neuronal GJs during synaptogenesis"

P21

"In particular, the fact that ZO1 localizes to sites of synaptic contact *independent of* the Connexins"

3. In Figure 5A and 5B, the size of the circle markers is too large. They mask other data points in the back and make data distribution unclear. Set the marker size smaller.

4. Figure S7 does not exist. It seems a typo: Figure S6 instead of S7.

5. P26: "The M-cell input resistance was estimated by applying a hyperpolarizing-current step of -1 nA and 20 ms in duration and measuring the voltage deflection caused, followed by derivation of resistance with Ohm's law."

Describe whether the authors subtracted pipette resistance from the calculated value.

6. Please include membrane potential recordings in all figures. Do any of the treatments substantially alter Vm? Why are membrane potentials missing everywhere?

7. The Discussion has a paragraph that is more than 2 pages. Please break that paragraph, and perhaps look at the Discussion and use headings to highlight what the authors think the essential issues are.

Reviewer #1:

In this manuscript, the authors use zebrafish Mauthner cells as a model to investigate the function of zo1 in gap junction regulation. They present solid evidence to show the role of zo1 in neuronal gap junction formation.

Overall, the data were clear and solid and show that zo1 is required for gap junction formation through binding with CX. However, the authors appear to conclude that zo1 also has a role in gap junction regulation/function in a similar fashion as that of scaffold proteins in chemical synapses, which is not well supported by any data. All function defects in zo1(b) mutants were likely originated from lack of gap junctions rather than failure in gap junction function. As motioned in the introduction part, gap junctions are "often perceived as simple aggregates of intercellular channels", and "it remains undetermined whether such associated proteins are ancillary to the channels or requisite for electrical synapse function". It will be very exciting if zo1 as a scaffold protein is involved in gap junction functions. Evidence presented in this study only showed that zo-1 is important for gap junction formation. Although zo1 has not been shown to be involved in neuronal gap junction formation, many studies in other cell types have shown the critical role of zo1 in gap junction formation.

Reviewer #2:

This paper reports the functional relationship between neuronal Connexins and Zonula Occludens 1 (ZO1). By performing chimera experiments, the authors elegantly showed that ZO1 is asymmetrically localized exclusively postsynaptically at neuronal contacts where it functions to assemble intercellular channels. The authors' findings challenge current perceptions of the functional and molecular organization of electrical synapses and calls for a new model that includes a primary role for the intracellular molecular scaffold in governing the function of intercellular channels.

---

## [Author Response]

Only the following revisions need to be implemented:1. The conclusion that ZO1 plays a role in gap junction regulation/function is not well supported by the data presented here as the authors cannot rule out that fact that gap junctions simply fail to form in ZO1 mutants. The authors should revise their conclusions in results and discussion to tone down the claim.

We appreciate the Reviewers’ concern and suggestions. We realize that we were not sufficiently clear when describing our perspective on the possible functional roles of ZO1 at Mauthner cell mixed synapses. Here we address the use of ‘function’ and ‘regulation’ in the text:

As stated in the first sentence of the Discussion section, we propose that ZO1 ‘is essential for the formation and function of electrical synapses’, not only function. Why? As pointed out by Reviewer 1, ZO1 could have a role in the formation of gap junctions. However, the existence of abortive synaptic structures in ZO1b-/- mutant fish (see Figure 1H-J) indicates that the neurons are attempting to construct and stabilize gap junction channels at the membrane, suggesting the existence of junctional structures. This finding raises the possibility that ZO1 could be functionally required for efficient insertion of channels at gap junctions. Such a possibility is consistent with previous data at these synapses (Flores et al. 2012) and with the proposed role of ZO1 at Cx43-containing gap junctions (Rhett and Gourdie, 2012). Finally, as a key scaffold, ZO1 might serve various functions, including both the formation and maintenance of functional channels, as well as potentially scaffolding other interactors such as kinases/phosphatases. Given all these possibilities, we cautiously chose to argue that ZO1 is likely to be involved in the ‘formation and function’ of electrical synapses.

We only discuss ‘regulation’ on within the Discussion, where we contrast differences in the role of ZO1 at Cx43 vs. Cx36-related gap junctions. In this section, we discuss potential regulatory roles of ZO1 at electrical synapses that were suggested in previous papers and tried to put our findings in the context of these speculations. However, we realize now that the last sentence of the paragraph could be seen as too strong and have now toned it down.

2. In Figure 2, the authors quantitated ZO1 localization to CEs or M/CoLo and showed that the amount of the localized ZO1 was halved in the connexin mutants. This means that the ZO1 localization to CEs or M/CoLo at least in part depends on connexins. This result should be carefully discussed in the Discussion section. The finding was described in the summarizing sentence on the page 8, "1) ZO1b localizes to putative electrical synaptic sites largely independent of Connexin proteins", but was not carefully described in the other sentences. The following sentences should be modified by weakening the asserting phrases in the main texts.P8"ZO1b localizes to the electrical synapse independent of Cxs""ZO1 does not require the presence of channel-forming proteins to localize at neuronal GJs during synaptogenesis"P21"In particular, the fact that ZO1 localizes to sites of synaptic contact independent of the Connexins"

We appreciate the suggestions and have clarified the indicated phrases. We have added an additional discussion of the reciprocal interaction to the Discussion.

3. In Figure 5A and 5B, the size of the circle markers is too large. They mask other data points in the back and make data distribution unclear. Set the marker size smaller.

Thank you, figure is updated.

4. Figure S7 does not exist. It seems a typo: Figure S6 instead of S7.

Thank you, all supplemental figure instances updated to *eLife* style.

5. P26: "The M-cell input resistance was estimated by applying a hyperpolarizing-current step of -1 nA and 20 ms in duration and measuring the voltage deflection caused, followed by derivation of resistance with Ohm's law."Describe whether the authors subtracted pipette resistance from the calculated value.

As described in the Results and Methods sections, electrophysiological recordings were performed under the current clamp configuration, which allows compensation of the electrode resistance with the ‘bridge’ balance. This compensation was done automatically using the feature provided by the 700A Multiclamp amplifier followed by manual adjustment, if needed, for accurate compensation. Reference to bridge compensation is now included in the methods section.

6. Please include membrane potential recordings in all figures. Do any of the treatments substantially alter Vm? Why are membrane potentials missing everywhere?

As requested, we now include the values of membrane potential for the illustrated recordings in their corresponding figure legends. They were not previously reported for stylistic reasons, following the custom of the Pereda laboratory. The averaged values of resting potential for wild type and mutant zebrafish are reported in Table 1.

Finally, bath application of MA significantly altered the resting potential of the Mauthner cells in wild type zebrafish, averaging -80.2 ± 0.7 mV in control and -84 ± 1 mV after MA application (p = 0.04, n = 5). In contrast, bath application of CNQX/DAP5 did not alter the resting potential of the Mauthner cell in wt, tjp1b -/-, gjd2a -/- or gjd1a -/- zebrafish. Following the reviewers’ suggestion, we have now added reference to these observations (see electrophysiology results section).

7. The Discussion has a paragraph that is more than 2 pages. Please break that paragraph, and perhaps look at the Discussion and use headings to highlight what the authors think the essential issues are.

Thank you for the suggestion, the Discussion has been updated with headings and edited for clarity/brevity.